# Bat IFITM3 restriction depends on S-palmitoylation and a polymorphic site within the CD225 domain

Camilla TO Benfield[1], Farrell MacKenzie[2], Markus Ritzefeld[3], Michela Mazzon[2], Stuart Weston[2], Edward W Tate[3], Boon Han Teo[1], Sarah E Smith[5], Paul Kellam[4,5], Edward C Holmes[6], Mark Marsh[2]

**Host interferon-induced transmembrane proteins (IFITMs) are broad-spectrum antiviral restriction factors. Of these, IFITM3 potently inhibits viruses that enter cells through acidic endosomes, many of which are zoonotic and emerging viruses with bats (order Chiroptera) as their natural hosts. We previously demonstrated that microbat IFITM3 is antiviral. Here, we show that bat IFITMs are characterized by strong adaptive evolution and identify a highly variable and functionally important site—codon 70—within the conserved CD225 domain of IFITMs. Mutation of this residue in microbat IFITM3 impairs restriction of representatives of four different virus families that enter cells via endosomes. This mutant shows altered subcellular localization and reduced S-palmitoylation, a phenotype copied by mutation of conserved cysteine residues in microbat IFITM3. Furthermore, we show that microbat IFITM3 is S-palmitoylated on cysteine residues C71, C72, and C105, mutation of each cysteine individually impairs virus restriction, and a triple C71A-C72A-C105A mutant loses all restriction activity, concomitant with subcellular re-localization of microbat IFITM3 to Golgi-associated sites. Thus, we propose that S-palmitoylation is critical for Chiropteran IFITM3 function and identify a key molecular determinant of IFITM3 S-palmitoylation.**

## Introduction

Interferon-induced transmembrane proteins (IFITMs) are antiviral factors that act uniquely and early in viral replication cycles to restrict the entry of a diverse range of primarily enveloped viruses into cells (1). Humans possess three IFN-inducible IFITM genes—*IFITM1*, *IFITM2*, and *IFITM3*—encoding proteins with antiviral functions and two IFITM family members that lack antiviral function—*IFITM5* and *IFITM10*. Mice have orthologs of all these IFITMs as well as two additional genes, *Ifitm6* and *Ifitm7*. Phylogenetic analysis of vertebrate IFITMs indicates that *IFITM1*, *IFITM2*, and *IFITM3* group with murine *ifitm6* and *ifitm7* in a clade of immunity-related IFITMs (IR-IFITMs), with *IFITM5* and *IFITM10* falling as separate lineages (2). IFITMs belong to the CD225/pfam04505 or "dispanin" protein superfamily (http://pfam.xfam.org/family/PF04505) (3) that contains more than 2,000 members, including both prokaryotic and eukaryotic proteins, all of which encode a conserved CD225 protein domain.

As their name suggests, IFITMs are membrane proteins, allowing them to police the cell surface and endocytic membranes that viruses must cross to invade cells. Studies of IFITM topology suggest a type II transmembrane configuration with a cytosolic N terminus, cytosolic conserved intracellular loop (CIL) domain, transmembrane domain, and extracellular (or intraluminal) C terminus (4, 5), although there is evidence that other IFITM topologies exist (6, 7, 8). The results of spectroscopic topological studies agree with the type II transmembrane configuration, as do bioinformatic predictions of IFITM3 secondary structure that reveal three alpha helices, with the C-terminal helix forming a single transmembrane domain (9, 10). The CD225 domain is highly conserved among IFITMs and comprises an intramembrane domain (IMD) and CIL domain. The hydrophobic IMD contains a 10-residue amphipathic helix (amino acid residues 59–68 of human IFITM3) that is required for the antiviral activity of both IFITM3 and IFITM1 (9). The subcellular localization of IFITMs is a key determinant of their antiviral profile. When expressed singly, IFITM3 and IFITM2 preferentially localize to early and late endosomes and lysosomes, restricting viruses that enter via these endolysosomal compartments. In contrast, IFITM1 primarily localizes at the cell surface and can restrict viruses that enter through the plasma membrane (11, 12, 13, 14). Indeed, mutants of IFITM3 that lack an N-terminal endocytic sorting motif [20]YEML[23] localize to the plasma membrane and lose their ability to inhibit influenza A virus (IAV), alphavirus, and coronavirus infection by endosomal routes (14, 15, 16, 17, 18).

[1]Department of Pathobiology and Population Sciences, Royal Veterinary College, University of London, London, UK [2]Medical Research Council Laboratory for Molecular Cell Biology, University College London, London, UK [3]Department of Chemistry, Imperial College London, London, UK [4]Department of Infectious Disease, Imperial College Faculty of Medicine, Wright Fleming Institute, St Mary's Campus, London, UK [5]Kymab Ltd, The Bennet Building (B930), Babraham Research Campus, Cambridge, UK [6]Marie Bashir Institute for Infectious Diseases and Biosecurity, Charles Perkins Centre, School of Life and Environmental Sciences and Sydney Medical School, The University of Sydney, Sydney, New South Wales, Australia

Correspondence: cbenfield@rvc.ac.uk
Stuart Weston's present address is Department of Microbiology and Immunology, University of Maryland School of Medicine, Baltimore, MD, USA

Studies focusing on IFITM3 restriction of IAV and Semliki Forest virus (SFV) indicate that virus internalization is unaffected by IFITM3 expression and, for SFV at least, the viral envelope glycoprotein undergoes low pH-induced conformational changes (14). However, for both viruses, the viral core components are not delivered to the cytoplasm, suggesting that membrane fusion fails. Experiments with IAV indicate that hemifusion (i.e., lipid-mixing between viral and cellular membranes) can occur in the presence of IFITM3, but the subsequent formation of a fusion pore is inhibited (13, 19). Recent work has shown that IFITM3-positive vesicles fuse with incoming virus-bearing vesicles before hemifusion and that IFITM3 enhances the rate of virus trafficking to lysosomes (20). The co-localization of viral cargo with IFITM3-positive endosomes is specific to restricted viruses, suggesting that IFITM-insensitive viruses such as Lassa virus enter via different endosomal compartments and thereby escape IFITM engagement and restriction (13, 20). Further examples of virus-specific IFITM action include the ability of murine IFITM6 to restrict filoviruses, but not IAV (21), and amino acids within the IFITM3 CIL domain that are preferentially needed for IAV but not dengue virus restriction (22). Other post-entry mechanisms for IFITM3 restriction have also been proposed (23, 24, 25).

IFITMs are heavily regulated by posttranslational modifications (PTMs). One major modification is S-palmitoylation, a reversible 16-carbon lipid PTM that increases protein hydrophobicity and influences the behavior of proteins in membrane environments (26). For human and murine IFITM3, S-palmitoylation can occur on cysteine residues 71, 72, and 105 and enhances IFITM3 antiviral activity (27, 28). Recent live-cell imaging showed that abrogating C72 palmitoylation slowed IFITM3 trafficking to membrane compartments containing IAV particles (20). Multiple zinc finger DHHC (Asp-His-His-Cys) domain–containing palmitoyltransferases (ZDHHCs) can palmitoylate IFITM3 with marked functional redundancy, although ZDHHC20 may be particularly important (29). For human IFITM3, C72 is also the dominant site for acylation (30). Three other PTMs have also been reported, all of which negatively regulate IFITM3 antiviral activity: ubiquitination on one or more of four lysine residues (27), methylation on K88 (31), and phosphorylation on Y20 (15, 16). IFITM3 also forms homo- and hetero-oligomers. Although these are thought to require amino acids F75 and F78 (22), a recent study reported that these residues are required for antiviral activity but not for IFITM3-IFITM3 interactions (32). In tissue culture systems at least, IFITM3, IFITM2, and IFITM1 restrict viruses in a cooperative manner, with the extent of cooperativity and redundancy between IFITMs varying for different viruses (20, 22). However, the biochemical mechanisms and molecular determinants by which IFITM proteins restrict virus infection are still far from clear.

In mice and humans, there is evidence that IFITM3 influences host antiviral resistance. *Ifitm3* knockout mice show enhanced morbidity after infection with IAV, alphaviruses, and flaviviruses (33, 34, 35, 36), and in humans, single nucleotide polymorphisms in *IFITM3*, which may act by altering IFITM3 expression or subcellular distribution, have been associated with an increase in morbidity in IAV and HIV-1 infections (recently reviewed by references 37 and 38). Antiviral function has also been reported for IFITMs from diverse mammalian and avian species (39, 40, 41, 42, 43, 44), although little is known about the role of IFITMs in antiviral defense in these species and in shaping host range. Phylogenetic studies have shown that the IFITM gene family is evolutionarily conserved and characterized by gene duplication, copy number variation, and the presence of pseudogenes (2, 45). Whereas humans possess a single *IFITM3* gene, a remarkable multiplicity of *IFITM3*-like genes exists in the genomes of some other primates; for example, there are 25 *IFITM3*-like variants in the marmoset and eight in the African green monkey (2, 46). Interestingly, several of these variants contain N-terminal polymorphisms which, when inserted into human IFITM3, prevent ubiquitination and endocytosis; however, the function of these duplicated *IFITM3*-like genes has not been tested (46).

The antiviral responses of bats (mammalian order Chiroptera) are of particular interest because these animals have been increasingly recognized as reservoir hosts from which numerous viruses have ultimately emerged, with severe pathogenic and socioeconomic consequences in humans and livestock. Indeed, a recent analysis showed that bats host a significantly higher proportion of zoonotic viruses than other mammalian orders (47). Moreover, Chiroptera are the only mammalian order that harbor significantly more zoonotic viruses than predicted from reporting effort and host traits such as geographic range and mammal sympatry. It is possible that this propensity to be reservoirs for a large number of viruses, many of which may remain asymptomatic, in part reflects aspects of bat immunology (48, 49, 50, 51, 52). Hence, the study of Chiropteran antiviral effectors, such as IFITMs, can potentially reveal mechanisms of viral tolerance as well as the evolutionary signatures of virus–host co-evolution. We previously showed that microbat IFITM3 retains sequence motifs for endocytosis and PTM, traffics to the plasma membrane before endocytic uptake, co-localizes with endosomal markers, and at normal expression levels in primary microbat cells, inhibits infection by pH-dependent enveloped viruses (39).

Here, we performed evolutionary analyses of mammalian IFITMs, including those identified from bats, to shed light on IFITM function and the nature of past selection pressures, and to identify key amino acids for experimental studies of IFITM function.

## Results

### Phylogenetic analysis of Chiropteran IFITM genes

Viral restriction activity is conserved in microbat IFITM3 (39). However, Chiropteran IFITMs have not been included in previous phylogenetic analyses, although bats comprise ~20% of all mammalian species. Accordingly, we first analyzed the phylogenetic relationships of IR-IFITM gene sequences from 31 eutherian mammal species, including 13 species of bat. The genes analyzed were identified either via translated BLAST (tBLAST) or from our cDNA analyses (rapid amplification of cDNA ends [RACE] on cells from *Myotis myotis*, *Eptesicus serotinus*, and *Sus scrofa* and proteomics informed by transcriptomics (53) for *Pteropus alecto*). The resulting phylogeny revealed relatively high levels of sequence divergence (nucleotide substitutions per site) between the IFITM genes (Fig 1). Notably, Chiropteran IFITMs formed a monophyletic group separated from other taxa by a relatively long branch.

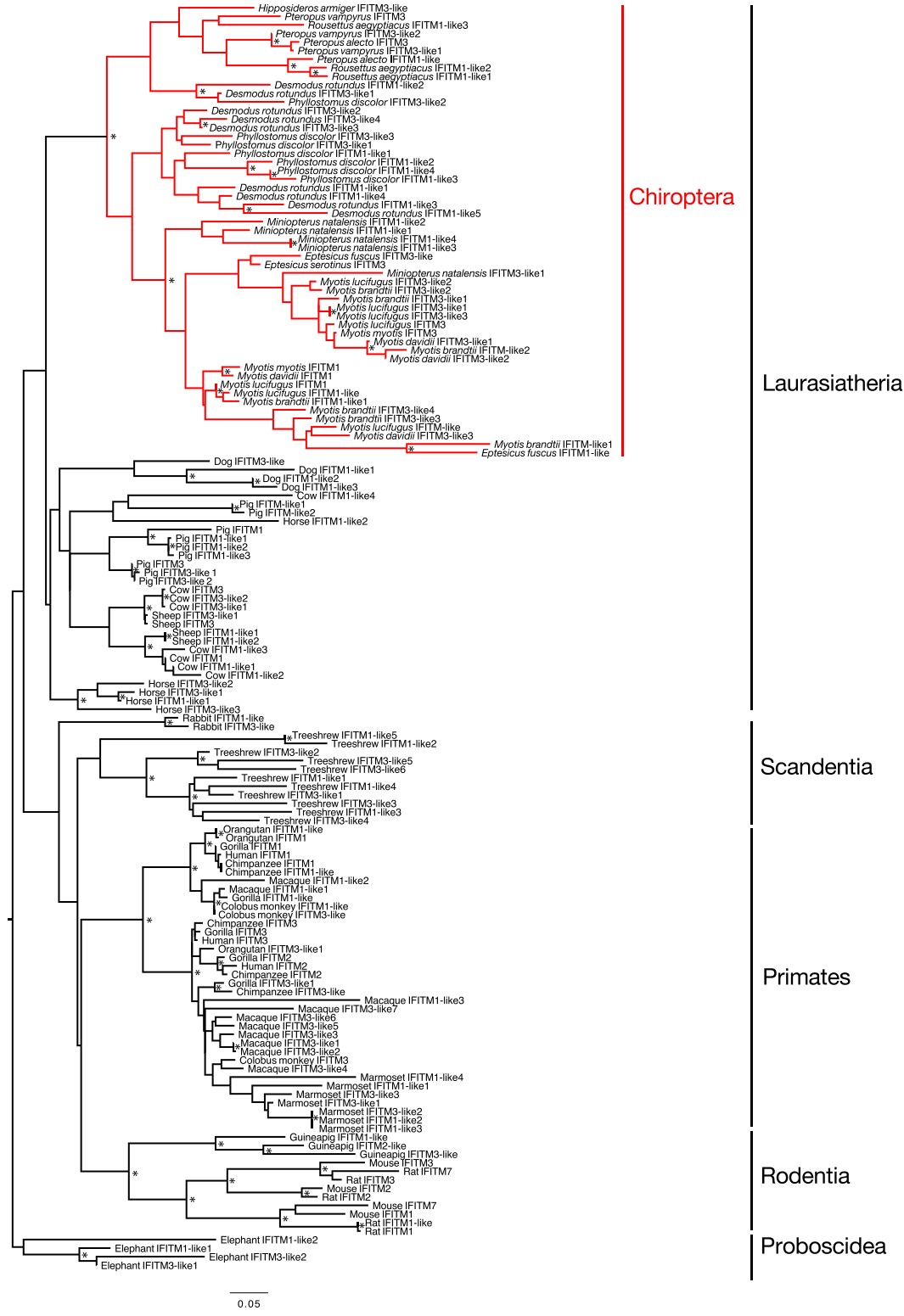

**Figure 1. Evolutionary history of mammalian IFITM genes.**
The phylogeny represents a maximum likelihood analysis of 148 IFITM genes, with mammalian orders and the Laurasiatheria clade indicated. The Chiropteran IFITM genes are shown in red. The tree was rooted according to the overall mammalian phylogeny (54). Nodes marked with * have both SH-support value > 0.9 and bootstrap value > 70%. Scale bar shows the number of nucleotide substitutions/site.

## Adaptive evolution on the branch leading to the Chiropteran IFITMs

The IFITM gene family is notable for gene duplications, variation in copy number, and the presence of pseudogenes (2, 55), and our previous tBLAST searches identified multiple IFITM-like genes for many species. In most cases, there are little data to differentiate functional from nonfunctional IFITM-like genes. Therefore, to assess possible selection pressures, we performed phylogenetic analysis on a selection of mammalian IR-IFITM genes for which there is evidence of function (Table S1): this data set comprises 31 IFITMs from 17 species, including eight Chiropteran species (Fig 2A). The selection pressures acting on these IFITM genes were assessed through analyses of the numbers of non-synonymous ($d_N$) to synonymous ($d_S$) nucleotide substitutions per site, with $d_N > d_S$ indicative of positive selection (i.e., adaptive evolution), performed using the adaptive Branch-Site Random Effects Likelihood method (56) that is able to determine selection pressures acting on specific branches of the input phylogeny. This analysis revealed that 6.8% of the codons in the sequence alignment were strongly positively selected on the branch leading to all Chiropteran IFITMs, with a $d_N > d_S$ of infinity (Likelihood ratio test = 14.95; $P$ = 0.002).

## Codon 70 is highly diverse among mammalian IFITMs

To visualize amino acid diversity, a logo diagram was constructed for the alignment of 31 "functional" IFITMs. This revealed notable amino acid diversity at codon 70, located within the otherwise highly conserved CD225 domain and directly upstream of the double cysteine motif C71/C72 (Fig 2B). The amino acids at codon 70 for the 31 functional mammalian IFITMs, as well the likely ancestral codons at this site at key nodes in the phylogeny inferred using a parsimony procedure, are indicated on Fig 2A. The human IFITM3, IFITM2, and IFITM1 proteins each encode a distinct amino acid at codon 70 (P, T, and W, respectively). A further three amino acids occur in IFITM proteins with confirmed antiviral activity: F (in mouse IFITM3 and IFITM1), A (in mouse IFITM2), and G (in mouse IFITM6; not used in the phylogenetic analysis). Assessing novel IFITM3 sequences, we obtained using RACE and RT-PCR for four species of microbat also highlighted codon 70 as the only non-conserved residue within the CD225 domain of microbat IFITM3, and it was notable that two microbat species encode valine (V) at this position (Fig 2C).

We next performed a site-specific selection analysis using the phylogeny of the 31 functional IFITMs shown in Fig 2A. Using the single-likelihood ancestor counting method, codon 121, within the C-terminal IFITM domain, was the only site identified as under significant positive selection ($P$ < 0.1; mean $d_N$-$d_S$ value of 2.64). Although there was no statistically significant evidence for positive selection on codon 70 ($P$ = 0.192), it had the second highest $d_N$-$d_S$ value of any site (1.91), with 14 non-synonymous nucleotide changes (compared with 13.5 at codon 121) and only two synonymous changes across the phylogeny. Similar results (i.e., close to significant positive selection) were obtained using the fast unconstrained Bayesian approximation method.

## Codon 70 affects antiviral restriction by microbat IFITM3

To assess the functional effects of variation at codon 70, polyclonal stable A549 cells were made expressing C-terminally HA-tagged wild-type (wt) human IFITM3 (i.e., huIFITM3 P70) and human IFITM3 encoding each of the other six amino acids that occur naturally in mammals at position 70 (i.e., A, T, G, W, F, and V). In addition, we made C-terminally HA-tagged wt microbat *M. myotis* IFITM3 (mbIFITM3) and a mbIFITM3 P70W mutant: W is a possible ancestral codon for the Laurasiatherian and primate IFITM clades (Fig 2A) and occurs in functional IFITMs such as pig IFITM3 and human IFITM1, but not in the bat IFITMs sequenced to date.

The mutant IFITM3 proteins were tested for their ability to restrict cell entry by Zika virus (ZIKV), SFV, and IAV (Fig 3). Flow cytometry was used to determine the proportion of infected cells after single cycle infection, and a representative experiment of three independent experiments is shown (Fig 3A–C). Each of the huIFITM3 codon 70 mutants restricted ZIKV infectivity with similar potency as wt huIFITM3 and wt mbIFITM3 (all <4% infection at an MOI of 10 and <1% infection at an MOI of 1). In contrast, mbIFITM3 P70W was significantly less restrictive than wt (35% ZIKV infection at an MOI of 10 and 7% at an MOI of 1) (Fig 3A). Similarly, small variations in infectivity of both SFV (Fig 3B) and IAV (Fig 3C) were apparent for the huIFITM3 codon 70 mutants, whereas infectivity of both viruses was markedly higher in cells expressing mbIFITM3 P70W compared with wt mbIFITM3. Expression levels of wt mbIFITM3 and mbIFITM3 P70W were comparable when analyzed by Western blotting (Figs 3A and B, S1A, and B) and flow cytometry (Fig S1C and D). Thus, differences in expression do not account for the different antiviral activities of these proteins. Using multi-cycle IAV infection, we investigated the impact on infectious yields and found that cells expressing mbIFITM3 P70W gave eightfold greater IAV titres than cells expressing wt mbIFITM3 (Fig 3D). Overall, the P70W substitution in mbIFITM3 significantly reduces restriction of representatives of three different virus families that enter cells via pH-dependent fusion from endosomes.

## P70W alters the subcellular localization of mbIFITM3

To assess the impact of P70 mutations on the subcellular localization of huIFITM3 and mbIFITM3, confocal immunofluorescence imaging was performed on A549 cells stably expressing either wt or P70W mutants of C-terminally HA-tagged huIFITM3 and mbIFITM3 (Fig 4). In fixed and permeabilized cells, wt huIFITM3 and mbIFITM3 had a punctate intracellular distribution, as described previously (5, 39), and huIFITM3 P70W showed a similar localization (Fig 4). In contrast, mbIFITM3 P70W showed a different distribution, with prominent perinuclear Golgi-like labelling (Fig 4A, permeabilized). When cells were labelled in the absence of detergent (Fig 4A, intact), some wt huIFITM3 was detected at the cell surface, as reported previously (5), and this was similar for huIFITM3 P70W. The strong cell surface labelling seen for wt mbIFITM3 was much reduced for mbIFITM3 P70W (Fig 4A, intact), suggesting reduced trafficking to the plasma membrane. To examine IFITM3 trafficking to the cell surface, the cells were incubated in medium containing anti-HA mAb at 37°C for 60 min before fixation with or without permeabilization (Fig 4B) to reveal internalised or cell surface anti-HA antibodies, respectively.

## A Phylogeny used for selection analyses

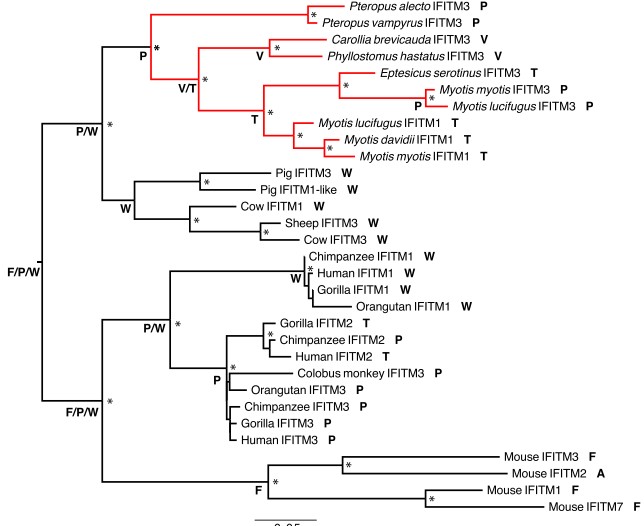

## B Sequence logo for functional IFITMs

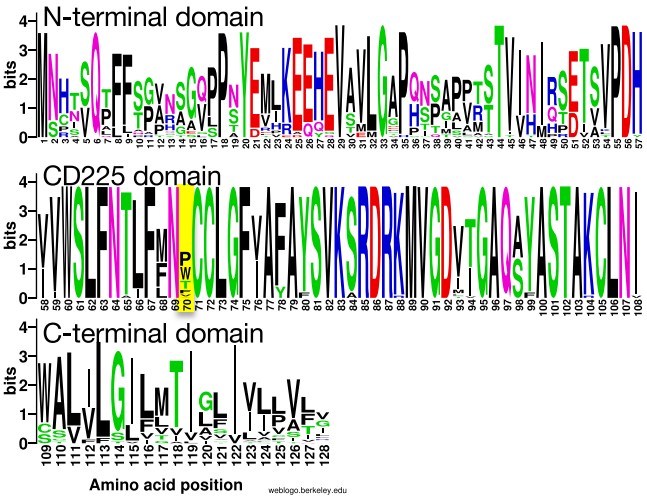

## C Alignment of IFITM3 from microbat species

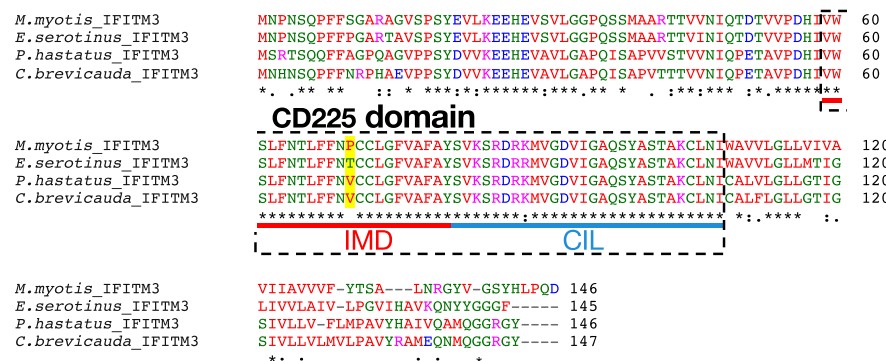

**Figure 2. Lineage-specific adaptive evolution in Chiropteran IFITMs and codon 70 diversity.**
**(A)** Phylogeny of 31 IFITM genes with evidence of function that was used to analyze selection pressures. A total of 384 well-aligned nucleotides was used for maximum likelihood phylogenetic analysis. The clade of Chiropteran IFITMs, identified as under strong positive selection, is indicated in red. Nodes marked with * have both SH-support value > 0.9 and bootstrap value > 70%. Scale bar shows the number of nucleotide substitutions/site, and the tree was rooted according to the overall mammalian phylogeny (54). The amino acid residues at codon 70 in mammalian IFITMs, as well as their likely ancestral states inferred using a parsimony procedure, are also shown. **(B)** Sequence logo for IFITM gene alignment used for Fig 2A with codon 70 highlighted. **(C)** Amino acid alignment of IFITM3 from four different microbat species. IFITM3

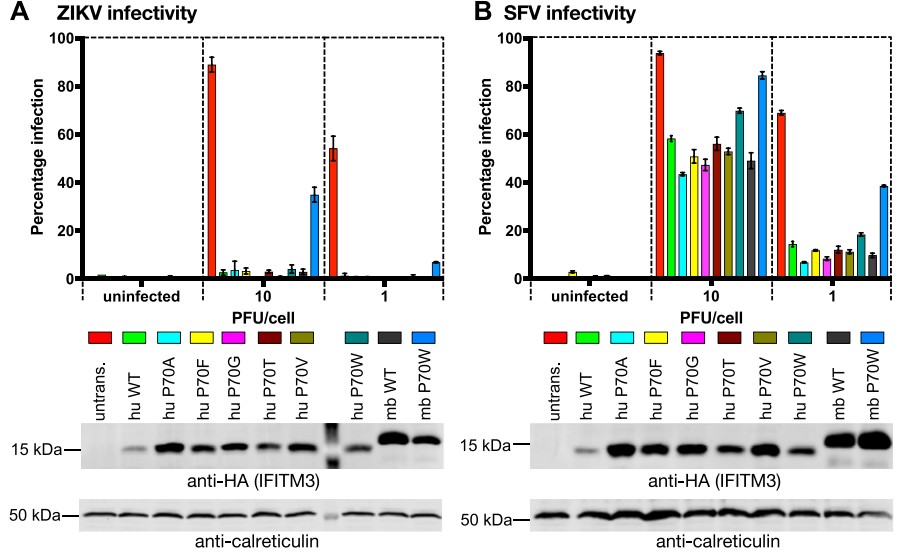

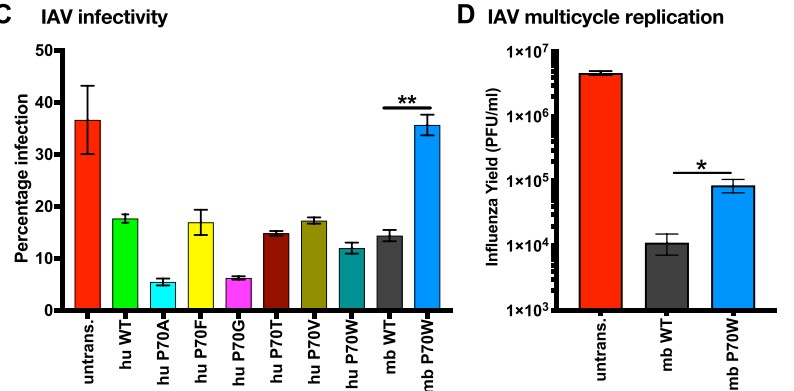

**Figure 3. Antiviral restriction by codon 70 mutants of human and microbat IFITM3.**
**(A, B)** A549 cells expressing the indicated IFITM3 proteins (or untransduced [untrans.] controls) were infected at an MOI of 10 or 1 (alongside uninfected controls) with either ZIKV for 28 h (A) or SFV_Zs-Green for 8 h (B). For (A), % infection was determined by flow cytometry using an anti-flavivirus primary antibody and AF 488–conjugated secondary antibody (% infection = % AF 488 positive cells). For (B), % infection was determined by flow cytometry as % of Zs-Green positive cells. **(A, B)** Data are from a representative experiment (n = 3) performed in triplicate (duplicate for uninfected controls) with each bar showing the mean ± SEM. Beneath each graph is a corresponding Western blot showing IFITM3 expression for that assay. IFITM3 proteins were detected with an anti-HA antibody. Calreticulin was used as a loading control. **(C)** A549 cells expressing the indicated IFITM3 proteins, or untransduced (untrans.) A549 cells, were infected with IAV at an MOI of 10. After 6 h, % infection (i.e., % NP positive cells) was determined using flow cytometry using FITC-conjugated anti-NP antibody. **(D)** Mean ± SEM of triplicate infections is shown (**P < 0.01, t test) (D) A549 cells expressing wt mbIFITM3 or mbIFITM3 P70W, or untransduced (untrans.) A549 cells, were infected with IAV at an MOI of 0.01 and 48 h later supernatants were harvested and the viral yields titrated on MDCK cells by plaque assay. Mean ± SEM of triplicate infections each titrated in duplicate is shown (*P < 0.05, t test).

In cells expressing huIFITM3s, similar intracellular punctate labelling was observed on permeabilized cells, suggesting that equivalent trafficking to the cell surface and subsequent endocytosis occurred for both wt and huIFITM3 P70W. Consistent with this, similar levels of cell surface labelling were seen for both proteins on intact cells. Significant intracellular labelling (permeabilized cells) and cell surface labelling (intact cells) was seen on cells expressing wt mbIFITM3, but little internal or cell surface staining was seen for cells expressing mbIFITM3 P70W. Together, these data indicate that the P70W mutation in mbIFITM3 significantly reduces trafficking of the protein to the cell surface and, as a consequence, there is little subsequent endocytic uptake of the protein into endosomal compartments.

As our initial observations suggested mbIFITM3 P70W may be preferentially associated with Golgi compartments (Fig 4A), we co-labelled fixed and permeabilized cells with anti-HA mAb (to detect the HA-tagged IFITM3 protein) and markers of intracellular organelles.

Clear overlap was seen for the perinuclear concentrations of mbI-FITM3 P70W with the *cis* Golgi marker Giantin (Fig 4C) and with the *trans* Golgi network marker TGN46 (Fig S2A), whereas the huIFITM3 proteins and wt mbIFITM3 showed significantly less overlap. Together, these data suggest that mbIFITM3 P70W showed increased association with the Golgi apparatus and reduced trafficking to the cell surface compared with the wt mbIFITM3 protein and the huIFITM3 proteins. Moreover, huIFITM3 P70W did not show any obvious alteration in its trafficking properties and behaved similarly to the wt protein. These properties correlated with the reduced viral restriction observed for mbIFITM3 P70W (Fig 3).

## P70W reduces S-palmitoylation of mbIFITM3

IFITM3 codon 70 is located directly adjacent to C71 and C72 (Fig 2C), which are conserved and S-palmitoylated in human and murine IFITM3. Therefore, we assessed S-palmitoylation levels of wt and

cDNA sequences were identified via RACE and RT-PCR (*M. myotis* previously (39)). Translated sequences were aligned using Clustal Omega. Text colour of amino acid residues denotes their physiochemical properties: blue, acidic; red, small and hydrophobic; magenta, basic; green, hydroxyl, sulfhydryl, amine, and glycine. An asterisk indicates identical amino acid residues, ":" indicates residues with strongly similar properties, and "." indicates residues with weakly similar properties. Codon 70 is highlighted in yellow. IFITM3 domains IMD and CIL, together comprising the CD225 domain, are shown (according to reference 22).

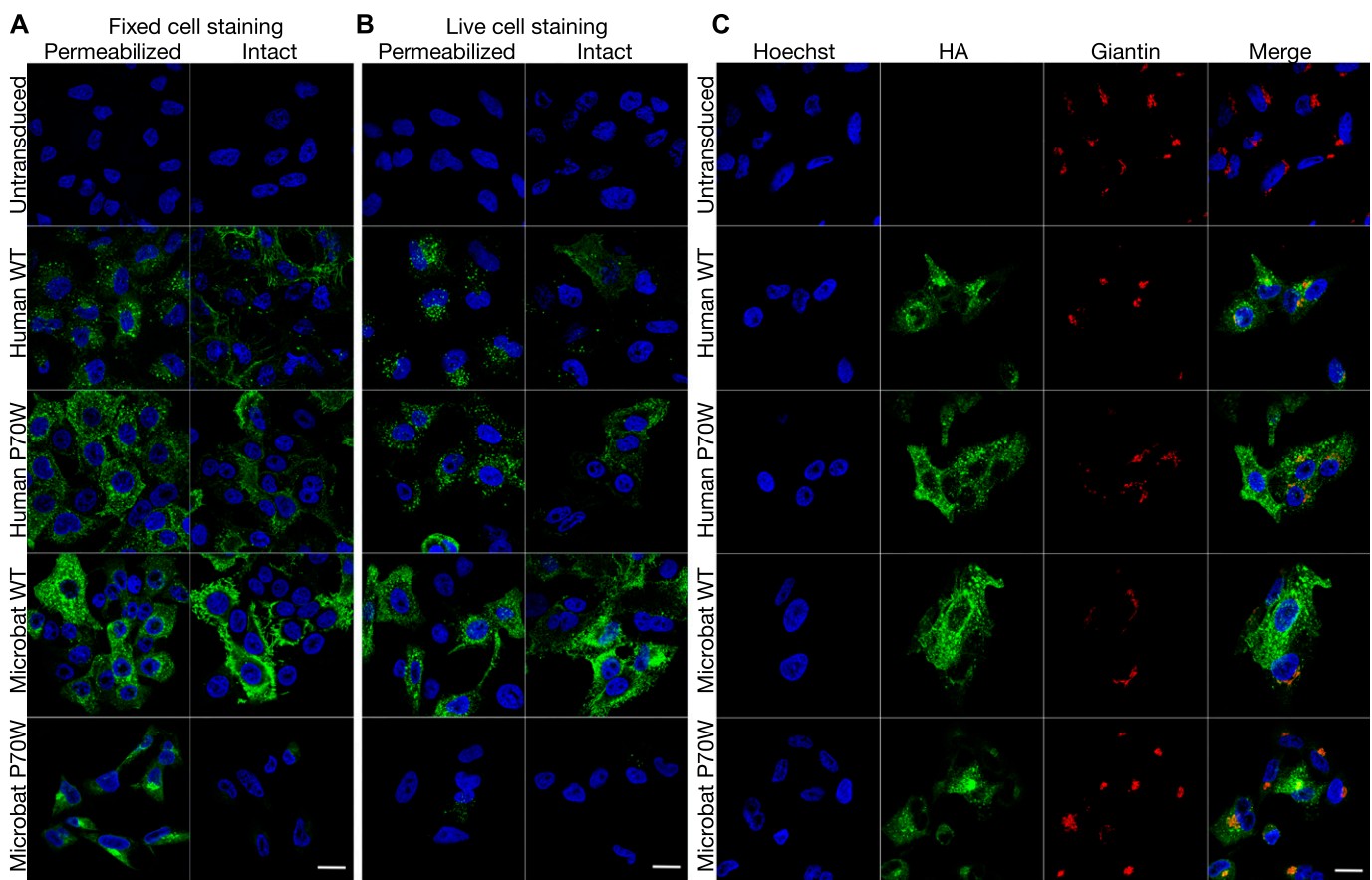

**Figure 4. Subcellular localization of IFITM3 proteins.**
Immunofluorescence of A549 cells stably expressing HA-tagged wt huIFITM3, huIFITM3 P70W, wt mbIFITM3, or mbIFITM3 P70W (or untransduced A549 cells). **(A)** Cells were fixed and stained for HA with permeabilization (permeabilized) or without (intact). **(B)** Live cells were incubated with anti-HA for 60 min at 37°C then fixed and subsequently stained with a secondary antibody with permeabilization (permeabilized) or without (intact). **(C)** Cells were fixed, permeabilized, and stained for HA and the *cis* Golgi marker Giantin. Nuclei were labelled with Hoechst and images taken on a confocal microscope. Scale bars represent 20 *μm*.

codon 70 mutants of hu and mbIFITM3 using a protocol based on the acyl-PEG exchange (APE) method developed by Percher et al (30, 57) in which we used maleimide-functionalised biotin to selectively substitute S-acylations (Fig S3A). Treatment of these samples with streptavidin induces a mass shift of the biotinylated proteins detectable by Western blotting (Fig S3B). Thereby, the intensity of the shifted band correlates with the amount of S-acylated protein. Whole cell lysates from IFITM3-expressing stable A549 cells were subject to the acyl exchange workflow (Fig S3A) and streptavidin treatment. After SDS–PAGE and anti-HA Western blotting, a band was observed that migrated at ~75 kDa (Fig 5B), which is consistent with the expected size of one streptavidin (Molecular Weight [MW] ~60 kDa) bound to one IFITM3-HA (MW ~15 kDa). Hence, this band represents S-palmitoylated IFITM3 present in the A549 cell lysates. The high MW band was dependent both on the presence of streptavidin and the addition of $NH_2OH$ (Fig S3B). Furthermore, titrating streptavidin showed that the amount used in this study was non-limiting (Fig S3B). Blurred bands at higher MW above the predominant 75 kDa band (Figs 5B and S3B) likely represent binding of additional streptavidin molecules to the second and/or third cysteine residue. Therefore, the entire area at and above the 75 kDa band was used for the densitometric analysis.

Triplicate lysates for each cell line were subjected to acyl exchange, independent SDS–PAGE, and anti-HA blotting. Densitometry was then performed in Image J to determine the proportion of each IFITM3 protein that was S-palmitoylated, that is, (HA signal ≥ 75 kDa)/(total HA signal in lane) (Fig 5A). The mbIFITM3 proteins showed a lower degree of S-palmitoylation compared with huIFITM3. Among the codon 70 mutants, the most marked difference was significantly reduced S-palmitoylation for mbIFITM3 P70W compared with wt mbIFITM3 (Fig 5A).

**Microbat IFITM3 is S-palmitoylated on C71, C72, and C105**

S-palmitoylation of IFITMs has only previously been studied for human and mouse proteins. To investigate S-palmitoylation in mbIFITM3, cysteine-to-alanine substitutions were made at conserved cysteine residues (C71, C72, and C105). A549 cells stably expressing single mutants C71A, C72A, and C105A, the double C71A-C72A mutant and the triple C71A-C72A-C105A mutant were generated. Two methods were used to assess S-palmitoylation: (i) the acyl exchange gel shift method described above (Figs 5, and 6A and B), and (ii) visualisation of palmitoylated proteins by in-gel fluorescence (Fig 6C and D). For this, the cells were metabolically labelled with a "clickable" alkyne

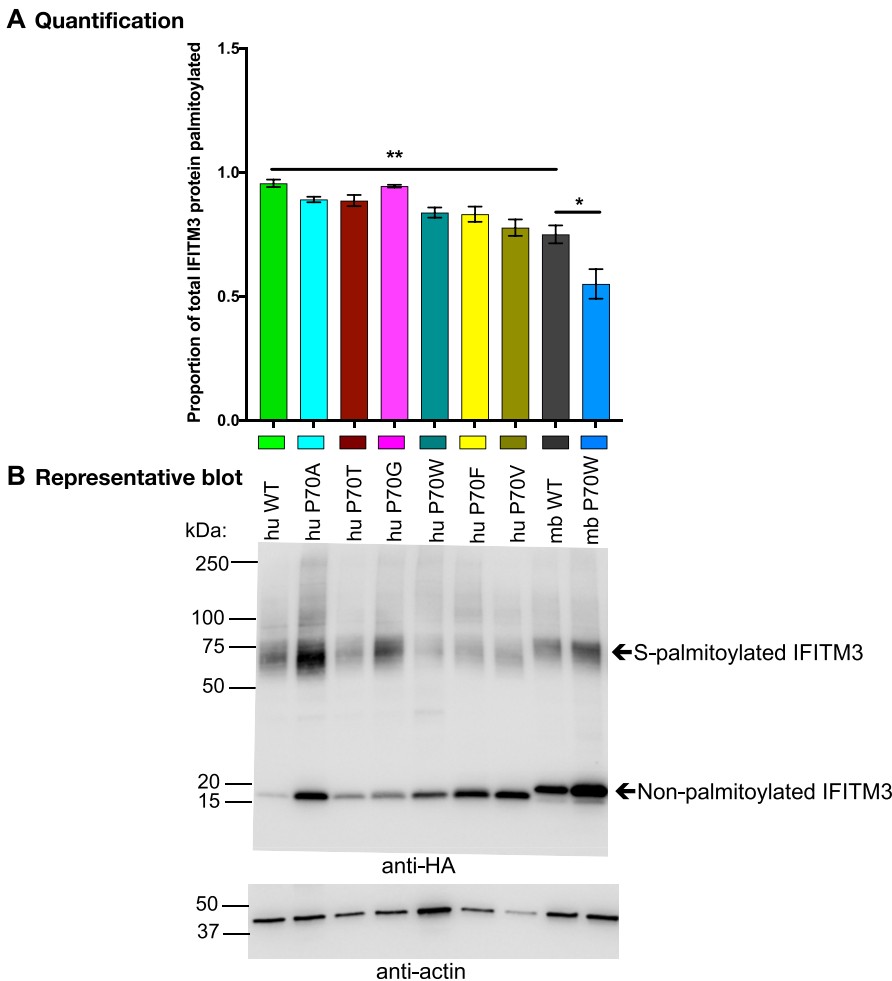

**A Quantification**

**B Representative blot**

**Figure 5.  S-palmitoylation levels of codon 70 IFITM3 mutants.**
A549 cells stably expressing the indicated HA-tagged hu and mbIFITM3 proteins were lysed and subjected to acyl–biotin exchange, in triplicate, for each IFITM3-expressing cell line. Streptavidin was then added to enable visualisation of biotin-modified (i.e., S-palmitoylated) IFITM3 by gel shift after SDS–PAGE (4 µg total protein loaded) and anti-HA Western blotting. **(A)** Proportion of total IFITM3 protein palmitoylated, given by densitometric quantification (Image J) of (shifted HA signal)/(total HA signal). Mean ± SEM of triplicate lysates each run on separate gels and blotted independently is shown (**P < 0.01, *P < 0.05, t test). **(B)** Representative anti-HA and anti-actin blot (same membrane). Arrows indicate the non-palmitoylated portion of IFITM3 running at ~15 kDa and the S-palmitoylated fraction which is shifted because of bound streptavidin.

palmitate analogue followed by further modification of the palmitoylated proteins via copper-catalyzed azide–alkyne cycloaddition reaction (i.e., "click chemistry") using a TAMRA-labelled capture reagent. The described metabolic labelling approach has been extensively used to successfully quantify a variety of PTMs (58, 59, 60). Comparing the two orthogonal methods (acyl exchange and metabolic labelling) used in this study reveals that both approaches have different informational value. The acyl exchange method only enables the quantification of the total amount of palmitoylated IFITM3. Because of the multi-valency of the streptavidin, multiple PTMs might still result in a 1:1 ratio of the IFITM3:streptavidin complex if the corresponding sites are in close vicinity. However, because of the significant shift of the IFITM3:streptavidin complex, palmitoylated and non-palmitoylated IFITM3 can be quantified in the same blot, thereby facilitating an internal normalization. The metabolic labelling approach enables the quantification of all palmitoylation sites, as each PTM contributes directly to the total TAMRA fluorescence intensity. However, for quantitative analysis, the TAMRA fluorescence must be normalized to the HA blot signal to account for variation in IFITM3-HA expression (Fig 6C) leading to larger error bars. The different informational value becomes obvious in the case of wt huIFITM3 and wt mbIFITM3. Although the TAMRA fluorescence for wt huIFITM3 was greater than for wt mbIFITM3 and

huIFITM3 P70W (Fig 6C), the acyl exchange result did not show a significant difference in the proportion of palmitoylated protein in either case (Fig 6A). Hence, the total amount of palmitoylated IFITM3 is equivalent in all three cases; however, the number of palmitoylations per protein differs.

Importantly, the results of both methods for the different cysteine mutants of mbIFITM3 showed that the single cysteine mutants (C71A, C72A, and C105A) have reduced palmitoylation compared with wt mbIFITM3, with a greater reduction for C72A than C71A. A double C71A–C72A mutant still showed some acylation, indicating that C105 can be acylated, whereas the triple mutant abrogated palmitoylation (Fig 6A–D). In summary, our data show that, as with human and mouse IFITM3, microbat IFITM3 can be S-palmitoylated on each of the three cysteine residues, with C72 being the preferred site. Furthermore, mbIFITM3 P70W showed a comparable reduction in S-palmitoylation to the single cysteine mutants of mbIFITM3.

## Palmitoylation-defective cysteine mutants of mbIFITM3 phenocopy the P70W mutation

We next studied the ability of the mbIFITM3 S-palmitoylation–defective cysteine mutants to restrict viral entry. The infectivity of ZIKV, SFV, and virus pseudotypes expressing the envelope proteins

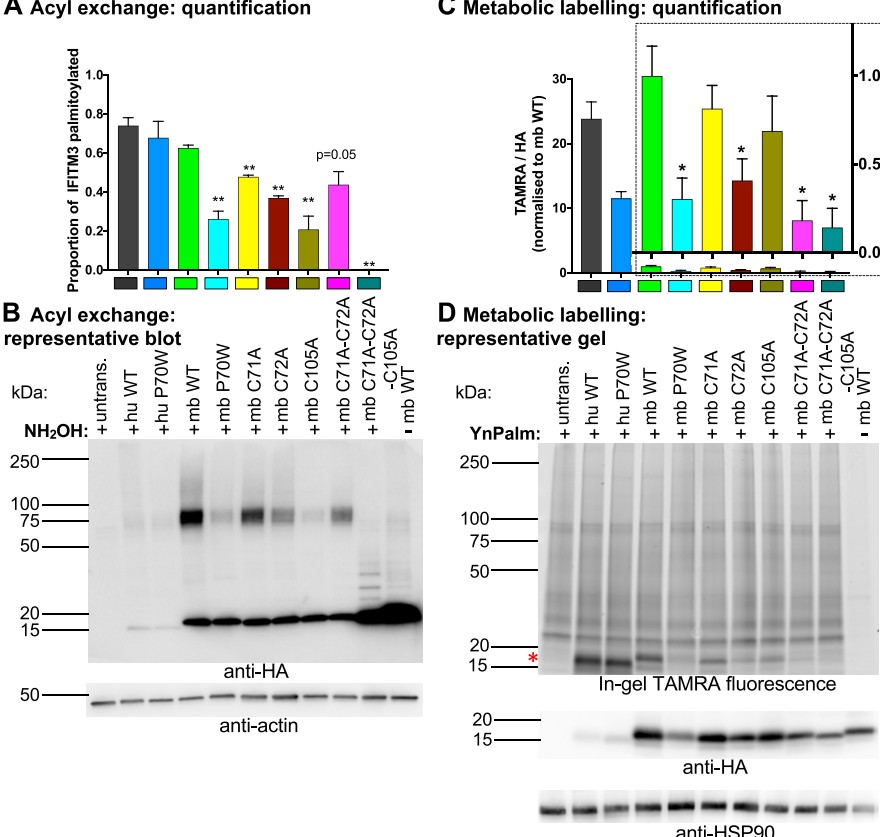

Figure 6.    S-palmitoylation levels of cysteine mutants of microbat IFITM3.

**(A, B)** A549 cells stably expressing the indicated HA-tagged hu and mbIFITM3 proteins, or untransduced (untrans.) A549 cells, were lysed, subjected to acyl–biotin exchange, and reacted with streptavidin, in triplicate for each IFITM3-expressing cell line. SDS–PAGE (4 µg total protein) and anti-HA and anti-actin Western blotting was then performed. **(A)** Quantification from triplicate acyl exchange experiments each run on separate gels and blotted independently (mean ± SEM) **P < 0.01 relative to wt mbIFITM3 (t test). The proportion of total IFITM3 protein palmitoylated is given by densitometric quantitation (Image J) of (shifted HA signal) /(total HA signal). **(B)** Representative Western blots for HA (IFITM3) and actin (loading control) performed on the same membrane. **(C, D)** A549 cells stably expressing the indicated HA-tagged IFITM3 proteins, or untransduced (untrans.) A549 cells, were incubated with the palmitate analogue YnPalm (at 50 µM) or with DMSO only (-YnPalm) for 16 h before lysis. Lysates were subjected to click chemistry using a TAMRA capture reagent before proteins were separated by SDS–PAGE (17 µg total protein loaded). Image J was used to quantitate the intensity of TAMRA (direct detection with Typhoon imager) and HA signal (from ECL) for each lane. **(C)** TAMRA/HA ratios, normalized to wt mbIFITM3, for triplicate YnPalm-treated lysates each subject to click chemistry, run on separate SDS–PAGE gels and blotted independently (mean ± SEM). *P < 0.05 relative to wt mbIFITM3 (t test). Inset bar chart in (C) shows TAMRA/HA for mbIFITM3 proteins, normalized to wt mbIFITM3, on an expanded scale. **(D)** A representative gel. The top panel shows in-gel TAMRA fluorescence, imaged directly with Typhoon imager. The red asterisk shows the position of IFITM3 proteins running at ~15 kDa. Lower panels show corresponding Western blots for the same gel for HA (IFITM3) and HSP90 (loading control).

of the lyssaviruses West Caucasian Bat virus (WCBV) and Lagos Bat virus (LBV) was tested in cells expressing either wt mbIFITM3, mbIFITM3 P70W, or the cysteine mutants of mbIFITM3 (Fig 7). All cysteine mutants showed impaired restriction relative to wt mbI-FITM3, with a predominant pattern of greatest restriction by C71A > C105A > P70W > C72A (Fig 7A–D). The double (C71A-C72A) and triple (C71A-C72A-C105A) mutants showed no restriction relative to control cells (Fig 7A–D). This is consistent with S-palmitoylation being greatest for C71A and abrogated for the triple cysteine mutant (Fig 6), indicating that palmitoylation positively regulates the antiviral activity of mbIFITM3. For comparison, wt huIFITM3 and huIFITM3 P70W were also tested in lyssavirus entry assays. As seen earlier for ZIKV, SFV, and IAV, the huIFITM3 P70W protein restricted lyssavirus entry as potently as wt huIFITM3 (Fig 7C and D).

Immunofluorescence analysis was performed on stable A549 cells expressing the HA-tagged versions of the different mbIFITM3 cysteine mutants. All single Cys mutants, as well as the double and triple mutants showed preferential perinuclear localization and a reduction in the more widely distributed punctate localization characteristic of wt mbIFITM3 (Fig 8A, permeabilized). The intra-cellular redistribution correlated with reduced exposure of IFITM3 at the cell surface (Fig 8A, intact). This was most marked for the double and triple cysteine mutants, which localized almost entirely to perinuclear sites and showed minimal cell surface staining above the background of untransduced cells. Of the mbIFITM3 point mutants, C72A had the most extensive perinuclear localization, correlating with it showing the greatest loss of restriction (Fig 7). The

perinuclear expression of the P70W mutant was as prominent as that seen for C105A and C71A, but for all three, a proportion of cells expressed some IFITM3 at the cell surface and more widely within the cells (Fig 8). As seen for mbIFITM3 P70W, the perinuclear IFITM3 co-localized with the *cis* Golgi marker Giantin (Fig 8B) and with the *trans* Golgi network marker TGN46 (Fig S2A), and to a much lesser extent with CD63-positive endolysosomes distributed throughout the cells (Fig S2B).

In sum, our results show that S-palmitoylation is necessary for viral restriction by microbat IFITM3 and that when S-palmitoylation is reduced, either directly by targeted cysteine mutation or in-directly via the P70W mutation, reduced restriction occurs con-comitantly with IFITM3 localization to perinuclear, Golgi-associated sites. Our data also indicate that for mbIFITM3 at least, defective S-palmitoylation and altered intracellular trafficking underlie the loss of viral restriction in the P70W mutant.

# Discussion

Chiroptera are the natural reservoir hosts for numerous zoonotic viruses, including IFITM3-sensitive viruses such as lyssaviruses and IAV, which may have an extended evolutionary history in bats (61, 62). IFITM3 is a critical component of antiviral immunity in vivo in humans and mice. However, its role in other species and its mechanism(s) of restriction is not fully understood. Comparative

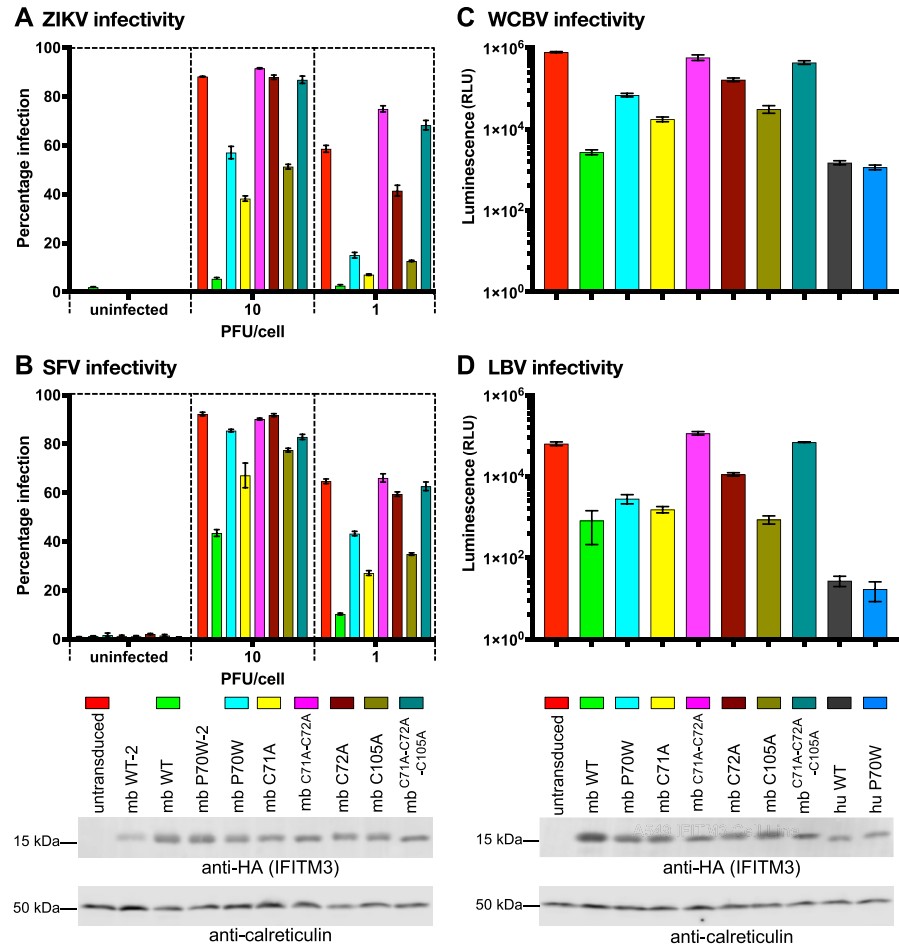

**Figure 7. Antiviral restriction by cysteine mutants of microbat IFITM3.**

**(A, B)** A549 cells expressing the indicated IFITM3 proteins, or untransduced controls, were infected at an MOI of 10 or 1 (alongside uninfected controls) with either ZIKV for 28 h (A) or SFV_Zs-Green for 8 h (B). Percentage infection was determined by high-throughput confocal imaging using Hoechst-stained nuclei to identify the total number of cells. For (A), the infected cells were detected with an anti-flavivirus primary antibody and AF 488–conjugated secondary antibody (% infection = % AF 488 positive cells). For (B), % infection = % Zs-Green positive cells. **(C, D)** For lyssavirus entry assays, A549 cells expressing the indicated IFITM3 proteins, or untransduced controls, were transduced with lentiviral pseudotypes expressing a firefly luciferase reporter and the G proteins from either WCBV (C) or LBV (D) and luciferase activity measured 48 h later (subtracting the luminescence values of uninfected controls to give the net luminescence values which were plotted on a log 10 scale). Each assay (A, B, C, and D) is a representative experiment (n = 3) performed in triplicate (duplicate for uninfected controls) with each bar showing the mean ± SEM. At the bottom of the figure are Western blots showing IFITM3 expression. IFITM3 proteins were detected with an anti-HA antibody and calreticulin was used as a loading control. **(A, B, C, D)** The blot on the left corresponds to (A) and (B), whereas the blot on the right corresponds to (C) and (D). The lanes labelled mb WT-2 and mb P70W-2 in the left blot refer to duplicate cell lines that were not used in the infectivity assays. RLU, relative light units.

inter-species genomic analysis is a powerful tool to understand host–pathogen interactions because genes and molecular determinants that function in host defense carry evolutionary signatures of adaptive evolution (63). Here, we included Chiroptera in evolutionary analyses of mammalian IR-IFITMs and identified strong adaptive evolution on the branch leading to bat IFITMs. Functional studies showed that codon 70, which exhibits extensive amino acid diversity among mammals, affects the antiviral activity and S-palmitoylation of mbIFITM3 and, furthermore, that S-palmitoylation is crucial for antiviral restriction by mbIFITM3 and has a marked effect on its subcellular localization.

Analysis of lineage-specific selection pressures revealed a strong signature of adaptive evolution on the branch leading to the Chiropteran IFITM clade. This is not a general feature of Chiropteran genes as, for example, the bat TLRs evolve under purifying selection (64), and evidence for positive selection is only seen in 1% of nuclear-encoded genes (65, 66). Importantly, the adaptive evolution we identified in bat IFITMs suggests that they serve an antiviral role in vivo, and that at least some bat viruses must have exerted major fitness costs on their hosts. Instances of virus-induced disease in bats have been reported (67, 68, 69), and the high viral richness per bat species (47) indicates that they harbor an exceptionally large number of viruses. Notably, Chiroptera seemingly harbor a greater variety of flaviviruses, bunyaviruses, and rhabdoviruses than any other mammalian order (47), all of which are RNA viruses restricted by IFITM proteins.

We previously showed that microbat IFITM3 is expressed constitutively in primary cells, as reported for IFITM3 orthologues in other species (33, 34, 41, 70, 71), and also induced by dsRNA, a pathogen-associated molecular pattern (39). It has been suggested that basal activation of the Chiropteran IFN system (72) limits viral replication and enables disease tolerance (73), and our findings indicate that bat IFITMs serve an adaptive function and may contribute to this mechanism. Interestingly, the evidence to date indicates that different species' IFITMs restrict viruses that are naturally hosted by these species; for example, duck IFITM3 against IAV (40), nonhuman primate IFITMs against SIV (43), and bat IFITM3 against lyssaviruses (39). This suggests that evolution of viral antagonism of IFITMs, although demonstrated for HIV-1 in vitro (74), may not occur readily in nature, perhaps because IFITMs act so early in the viral replication cycle.

Codon 70 is especially diverse across the phylogeny of 31 mammalian IFITMs and lies within the highly conserved IFITM CD225 domain adjacent to known functionally important residues C71/C72. We, therefore, chose to investigate the effects of amino acid diversity at this site and, to do so, substituted P70 in human IFITM3 with each of the other residues which occur naturally in functional mammalian IFITMs (A, T, G, W, F, and V). None of these mutants differed from wt in restricting three different viruses IAV, SFV, and

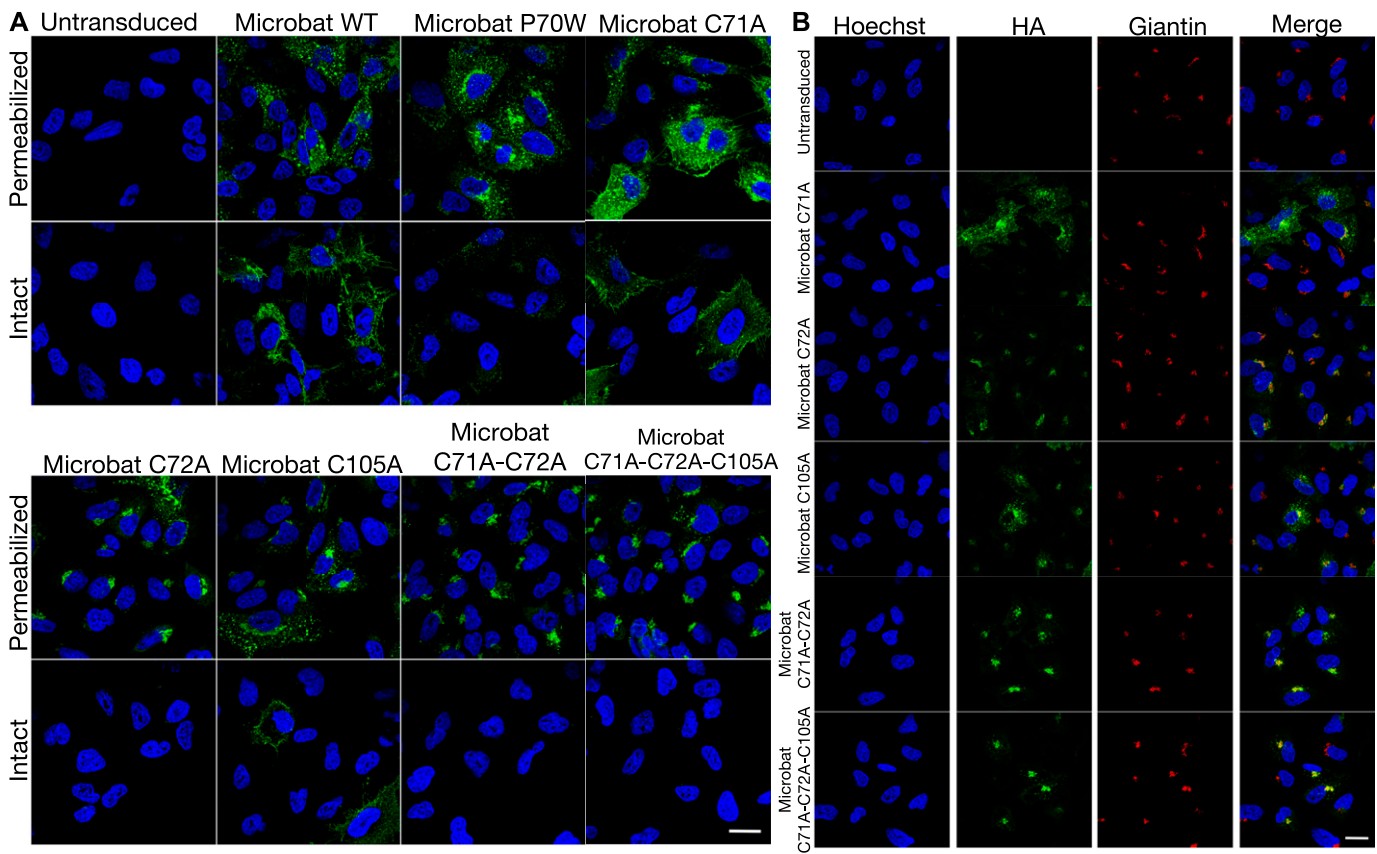

**Figure 8. Subcellular localization of cysteine mutants of microbat IFITM3.**
Immunofluorescence of A549 cells stably expressing wt HA-tagged mbIFITM3, mbIFITM3 P70W, mbIFITM3 C71A, mbIFITM3 C72A, mbIFITM3 C105A, mbIFITM3 C71A-C72A, or mbIFITM3 C71A-C72A-C105A (or untransduced A549 cells). **(A)** Cells were fixed and stained for HA with permeabilization (permeabilized) or without (intact). **(B)** Cells were fixed, permeabilized, and stained for HA and the *cis* Golgi marker Giantin. Nuclei were labelled with Hoechst and images taken on a confocal microscope. Scale bars represent 20 μm.

ZIKV. An earlier report also showed that substituting P70 of human IFITM3 with T, the residue present in human IFITM2, did not alter IAV restriction or subcellular localization (22). Our study isolated the effects of six different codon 70 residues within the background of human IFITM3 but does not exclude the possibility that these residues might have functional consequences in the context of different IFITMs or against other viruses.

To investigate the function of codon 70 in Chiropteran IFITMs, site-directed mutagenesis was performed on IFITM3 of the microbat *M. myotis* because this is the only Chiropteran IFITM reported to act as a restriction factor (39). A P70W substitution was made because tryptophan is common among the functional mammalian IFITMs in our phylogenetic analysis and was inferred as a possible ancestral codon (although the ancestral state reconstruction was ambiguous at the tree root). Microbat P70W showed reduced restriction of IAV, SFV, (*Alphaviridae*), ZIKV (*Flaviviridae*), and two lyssaviruses, concomitant with re-localization to perinuclear Golgi-associated sites and reduced S-palmitoylation. As for mbIFITM3 P70W, S-palmitoylation–defective cysteine mutants of microbat IFITM3 showed markedly different localization from wt, with reduced expression at the cell surface and retention at Golgi-associated sites. Based on the evidence that restriction relies on the localization of IFITM proteins at sites of viral entry (13, 14, 20), altered localization of the mbIFITM3

P70W and cysteine mbIFITM3 mutants likely explains their reduced restriction of viruses that enter through endosomes. However, whether deficient S-palmitoylation leads to altered localization or whether altered localization prevents the proper enzymatic processes of microbat IFITM3 S-palmitoylation remains to be studied. The 23 different human ZDHHC palmitoyltransferases localize to various different cellular sites, including the early secretory pathway (ER and Golgi apparatus), plasma membrane, and endocytic vesicles (75). McMichael et al have suggested that the location at which IFITM3 is palmitoylated may influence its activity because they found that ZDHHC20, which had the greatest effect on IFITM3 S-palmitoylation, also uniquely co-localized with IFITM3 at lysosomes (29).

Previous studies of equivalent S-palmitoylation–deficient cysteine mutants of human and murine IFITM3, including the triple C71A-C72A-C105A mutant, showed no change in IFITM3 subcellular localization (28, 30), whereas another study reported no change for C71A and C105A but a more centralized intracellular distribution for C72A than wt huIFITM3 (22). In contrast, all cysteine mutants of mbIFITM3 relocalised to perinuclear Golgi-associated sites. Thus, S-palmitoylation may positively regulate the antiviral activity of human, murine, and microbat IFITM3 via different mechanisms, including effects on protein–protein interactions, IFITM3 conformation,

membrane tethering/targeting, or the interplay with other PTMs (26, 30). As for human and murine IFITM3 (30), our data indicate that C72 is the major site of S-palmitoylation in microbat IFITM3 with C72A showing the most profound defect in restriction. Single mutation of each of C71, C72, and C105 reproducibly impaired restriction by microbat IFITM3 in stable A549 cells, indicating that each of these S-palmitoylation sites contributes to antiviral function. In contrast, C105A but not C71A, reduced restriction by murine IFITM3 (30), and neither C105A nor C71A significantly impaired the antiviral activity of human IFITM3 (22, 30), including when human IFITM3 mutants were stably expressed in A549s (22), as in the current study.

These qualitative differences, either in the mechanisms or site-specific effects of S-palmitoylation on human and microbat IFITM3, may explain why the P70W mutation in human IFITM3 did not alter its phenotype. Alternatively, quantitative differences may be responsible: huIFITM3 P70W was twofold less S-palmitoylated in our metabolic labelling experiments than wt huIFITM3, but still more S-palmitoylated than wt mbIFITM3. More S-palmitoylations on human compared with mbIFITM3 might enable wt huIFITM3 to tolerate some reduced S-palmitoylation (e.g., with P70W), perhaps suggesting a threshold level of S-palmitoylated IFITM3 is needed for restriction.

Codon 70 is located within a short unstructured portion of the IMD, [69]NPCC[72] in human and microbat IFITM3, which separates two alpha helices (9). The S-palmitoylated C71 and C72 residues are highly conserved among IFITMs as well as the larger dispanin protein superfamily (3). The other residue flanking codon 70, N69, is also strongly evolutionarily conserved in IFITMs (Fig 2B) and thought to be specifically required for proper termination of the amphipathic helix (9). It is unlikely that the phenotype of our codon 70 mutant relates to disruption of the amphipathic helix comprising residues 59–68 because mutations in IFITM3 that destroy the amphipathic helix, including complete deletion of the helix, are S-palmitoylated at wild-type levels and localize normally (9). The residue at codon 70 might affect the accessibility of C71/C72 for acylation, or its cytoplasmic facing position could enable interactions with ZDHHCs, with themselves, transmembrane proteins with cytosolic DHHC-containing catalytic domains, or with protein complexes such as those needed for ZDHHC-mediated S-palmitoylation of Ras (76). However, the recognition and specificity determinants of ZDHHC protein substrates remains poorly understood (77) and not yet studied in the case of IFITMs. Interestingly, codon 70 lies within a short stretch of amino acid residues which is only present in the vertebrate DSPA proteins and no other dispanin superfamily member (3), consistent with a role mapping to the antiviral function that resides uniquely in some DSPA members.

Why such diversity has evolved at codon 70 among mammalian IFITMs is unclear. It may reflect the diversity in IFITM-modifying ZDHHCs among different species or differentially regulated S-palmitoylation of different IR-IFITMs in the same species to ensure successfully coordinated antiviral action in host cells. Alternatively, diversity at codon 70 may have evolved because of differences in the magnitude or the type of virus-induced selection acting on different hosts. Addressing these possibilities will require a range of different residues to be systematically tested in their cognate IR-IFITMs and host cell background against a range of viral targets, and for the pleotropic cellular effects of IFITM expression to be similarly analyzed (25). Our initial studies show that huIFITM3 or mbIFITM3 wt or codon 70

mutants do not affect proliferation of A549 cells (Fig S4). The literature on the viral and cellular roles of IFITMs has remained largely discrete, although it is important to consider how these roles co-exist and are regulated, particularly in light of the therapeutic potential of IFITMs as antivirals. The recent report that IFITM expression inhibits placental and fetal development by preventing trophoblast cell fusion highlights a novel, noninfectious, selective pressure that may oppose that of antiviral defense to shape IFITM evolution and regulation (78, 79). Such reproduction-based selection pressure likely varies between mammals according to species' fecundity, life history traits, and placental physiology and, interestingly, may be another driver of variation among mammalian IFITMs.

# Materials and Methods

## Identification of IFITM genes

IFITM genes from 31 mammalian species were used for phylogenetic analysis (Fig 1), including three species of megabat and 10 species of microbat. IFITM sequences were derived either from our transcriptional analyses or by tBLAST searches of NCBI nucleotide and EST databases using human IFITM3 and IFITM1 as queries. For inclusion in phylogenetic analysis, genes identified via BLAST had to have a CD225 domain, intact ORF of ~IFITM size, and two exons. IFITM genes were cloned from primary fibroblasts from the microbat species *M. myotis*, the FLN-R cell line derived from microbat *E. serotinus* (kindly provided by Prof Martin Beer, Friedrich Loeffler Institute), and the pig NPTr cell line, using RACE and RT-PCR methods as previously described (39). The *P. alecto* IFITM3 gene sequence used was found using proteomics informed by transcriptomics (53), which provided evidence for transcript and protein expression in *P. alecto* cells. Uncharacterized IFITM sequences were named as "IFITM3-like" if their initiation codons aligned with known IFITM3 genes and "IFITM1-like" if their start codon aligned with other IFITM1 genes. Table S1 shows IR-IFITM genes with evidence for function (21, 39, 42, 46, 80) which were used to analyse selection pressures.

## Phylogenetic analysis

Multiple sequence alignment was performed using multiple runs of the MAFFT procedure (81) at different stringencies as available in the Geneious package (82). Sequence alignments were constructed for (i) 148 mammalian IFITM genes (alignment = 347 nt in length) and (ii) 31 functional IFITM genes (alignment = 384 nt in length). In the case of the 148 gene sequence alignment, all unaligned and ambiguously aligned regions were removed using Gblocks (83, 84), leaving 347 nt for analysis. Alignments are available from the authors on request. Maximum likelihood phylogenies of data sets were inferred using PhyML (85), employing the general time reversible model of nucleotide substitution and a gamma (Γ) model of among-site rate variation with a proportion of invariant sites (I).

## Analysis of clade-specific and codon-specific selection pressures

To assess selection pressures acting on IFITM genes (Table S1), we computed the relative numbers of non-synonymous ($d_N$) and

synonymous ($d_S$) nucleotide substitutions per site (i.e., the ratio $d_N/d_S$) among the 31 functional IFITM genes using both the single-likelihood ancestor counting and fast unconstrained Bayesian approximation programs as implemented in the Datamonkey web server (http://datamonkey.org), using default settings (86, 87). To assess the nature of selection pressures on the branch leading to the Chiropteran IFITMs we used the adaptive Branch-Site Random Effects Likelihood model available on Datamonkey (56).

## Cloning of microbat IFITM3 genes

RACE was conducted as described previously (39) on primary cells from *Pteropus hastatus* and *Carollia brevicauda*. Based on RACE-derived sequence, the following primers were designed to amplify full IFITM3 genes by PCR: *P. hastatus:* 5′-AGGAATCCGCTCTGTGTAGGG-3′ and 5′-GGGAAGGGTGACAGCCTCAGG-3′; *C. brevicauda* 5′-CATCTGCTCTGTTTAGGGACC-3′ and 5′-GCAGCCACCAGAAGCCTCCTA-3′. PCR performed on genomic DNA confirmed the presence of an intron. Genes were assigned as IFITM3 based on having a long N-terminus and double phenylalanine motif (F8/F9), as previously (39).

## Cells and viruses

A549 cells (ATCC CCL-185) were cultured in Ham's F-12 Nutrient Mixture with GlutaMAX (F12; Gibco), supplemented with 10% (vol/vol) FBS (Gibco). SFV_Zs-Green (a kind gift of Giuseppe Balistreri, University of Helsinki, described in reference 88) was expanded and titrated by plaque assay on BHK-21 cells (89). Zika virus (ZIKV) was expanded and titrated by immunofocus assay on Vero cells. Influenza A/WSN/33 virus (IAV) (H1N1) was grown in eggs and titrated by plaque assay on MDCK cells.

## IFITM3 mutagenesis

To generate mbIFITM3 P70W and huIFITM3 codon 70 mutants, C-terminally HA-tagged wt IFITM3 genes in the lentiviral vector pHR-SIN cGGW PGK puro were mutagenized using the QuikChange II XL Site-Directed Mutagenesis Kit (Agilent) according to the manufacturer's instructions using the following primers (all listed 5′ to 3′): mbIFITM3 P70W: CGAAGCCCAGGCAGCACCAGTTGAAGAACAGGGTGTTG and CAACACCCTGTTCTTCAACTGGTGCTGCCTGGGCTTCG; huIFITM3 P70A: TTCAACACCCTGTTCATGAATGCCTGCTGCCTG and CAGGCAGCAGGCATT-CATGAACAGGGTGTTGAA; huIFITM3 P70T: TGTTCAACACCCTGTTCATGAA-TACCTGCTGCCTGG and CCAGGCAGCAGGTATTCATGAACAGGGTGTTGAACA; huIFITM3 P70G: GTTCAACACCCTGTTCATGAATGGCTGCTGCCTGGG and CCCAGGCAGCAGCCATTCATGAACAGGGTGTTGAAC; huIFITM3 P70W: CCTGTTCAACACCCTGTTCATGAATTGGTGCTGCCTGGGCT and AGCCCAGG-CAGCACCAATTCATGAACAGGGTGTTGAACAGG; huIFITM3 P70F: CTGTTCAACACCCTGTTCATGAATTTCTGCTGCCTGGGC and GCCCAGG-CAGCAGAAATTCATGAACAGGGTGTTGAACAG; huIFITM3 P70V: GTTCAACACCCTGTTCATGAATGTCTGCTGCCTGGG and CCCAGGCAGCA-GACATTCATGAACAGGGTGTTGAAC. To generate cysteine mutants, C-terminally HA-tagged wt mbIFITM3 was subcloned from pHR-SIN CSGW PGK Puro into pcDNA3.1 by BamHI/NotI digest. Cysteine-to-alanine mutations were then introduced using the Q5 Site-Directed Mutagenesis Kit (NEB) according to the manufacturer's instructions. Primers were as follows (all listed 5′ to 3′): For C71A –

CTTCAACCCCGCCTGCCTGGGCTTC and AACAGGGTGTTGAACAGG; for C72A – CAACCCCTGCGCCCTGGGCTTCG and AAGAACAGGGTGTTGAACAG; for C105A CACCGCCAAGGCCCTGAACATCTGGGC and CTGGCGTAGCTCTGGGCG; for C71A-C72A – CTTCAACCCCGCCGCCCTGGGCTTCG and AACAGGGTGTTGAA-CAGG. The C71A-C72A-C105A construct was cloned using the C105A primers with the C71A-C72A construct as a template. Each of the five mutant sequences were then subcloned back into pHR-SIN CSGW PGK Puro by BamHI/NotI digest. All constructs were confirmed by Sanger sequencing.

## Generation of stable cell lines

A549 cell lines stably expressing variants of IFITM3 were generated by lentiviral transduction as previously described (39), but using the lentiviral vector pHR-SIN CSGW PGK Puro expressing C-terminally HA-tagged IFITM3 constructs. Transduction efficiency was checked after 48 h by flow cytometric detection of HA. Transduced cells were selected using 1.4 µg/ml puromycin (Thermo Fisher Scientific) and the resultant polyclonal cell lines used for this study.

## Infectivity assays

### ZIKV and SFV
Cells were infected in F12 with 2% FBS (infection media) and analyzed after 8 h for SFV and 28 h for Zika virus. For flow cytometric analysis, cells were trypsinized and fixed in 4% (wt/vol) formaldehyde (TAAB) for 20 min. For Zs-Green–expressing SFV infections, fixed cells were washed in PBS, resuspended in FACS resuspension buffer (1% [vol/vol] FBS, 2 mM EDTA in PBS), and stored at 4°C until analyzed. For ZIKV infections, fixed cells were washed in PBS, resuspended in FACS wash buffer (1% [vol/vol] FBS, 2 mM EDTA, 0.1% [wt/vol] saponin in PBS) for 25 min at RT, and then stained with mouse anti-flavivirus antibody (clone D1-4G2-4-15; Millipore) diluted 1:500 in FACS wash buffer for 90 min at RT. After washing twice with FACS wash buffer, the cells were incubated with Alexa Fluor (AF) 488–conjugated antimouse secondary antibody (diluted 1:800 in FACS wash buffer) for 1 h at RT. The cells were then washed twice in FACS wash buffer and stored in FACS resuspension buffer at 4°C until analyzed. For flow cytometry (LSR-II; BD Bioscience) cells were gated on forward and side scatter and analyzed for FITC fluorescence. The data were processed using FlowJo vX.0.7 software (Tree Star). For SFV and ZIKV infections analyzed by high-throughput confocal imaging (90), the cells were plated for infection in 96-well CellCarrier Ultra plates (Perkin Elmer). Fixation and staining conditions were as described for flow cytometry, but with nuclei stained with 5 µg/ml Hoechst-33258 (Sigma-Aldrich). Imaging was performed on an Opera LX system (Perkin Elmer). For each well, three images were acquired using a 4× objective (together accounting for one-third of the total well area). Total cell numbers and Zs-Green–infected cells or AF 488–positive cells were counted using the PE Columbus software (Columbus 2.4.1.110801, PE, UK).

### IAV
Analysis of influenza virus infectivity by flow cytometric detection of nucleoprotein (NP) and analysis of multi-cycle influenza replication was performed as described previously (39).

### Lyssavirus

Entry assays using lentiviral pseudotypes expressing a firefly luciferase reporter and the G proteins from either WCBV (AAR03484) or LBV (LBV.NIG56-RV1; HM623779) were described previously (39). Transductions were performed in triplicate on A549 cells in 96-well plates, and 48 h later, the cells were lysed with 5× Passive Lysis Buffer (Promega) and then added to luciferase substrate and luciferase activity measured on an Orion L Microplate Luminometer (Berthold Detection Systems).

### Immunofluorescence

Immunofluorescence on saponin-permeabilized cells was carried out as described previously (5) except that cells were fixed in methanol-free 4% (wt/vol) PFA (Alfa Aesar) in PBS. This same procedure was followed for intact cells except that saponin was omitted from the permeabilization buffer. For live cell staining, cells cultured on coverslips were incubated in media containing primary antibody for 1 h at 37°C with 5% $CO_2$. The cells were washed twice in the media and once in PBS to remove unbound antibody and then fixed and either permeabilized or left intact and processed as described above. Rat anti-HA primary antibody (clone 3F10, 1:100; Roche) was detected with either goat antirat conjugated to AF 488 (1:500) or with donkey anti-rat AF 488 (1:500). Rabbit anti-giantin primary antibody (discontinued, 1:1,000; Abcam) was detected with goat antirabbit AF 647, sheep anti-TGN46 primary antibody (AHP500, 1:100; Bio-Rad) was detected with donkey antigoat AF 647 (1:500) and mouse anti-CD63 primary antibody ((91); 1:10,000) was detected with goat anti-mouse AF 647 (1:500).

### Metabolic labeling and click chemistry analysis of S-palmitoylated proteins

A549 cells ($3 \times 10^5$ cells/well in a six-well plate) were incubated for 16 h with the alkyne palmitate (C16) analogue YnPalm (at 50 $\mu$M from 50 mM stock dissolved in DMSO) in F12 media with 3% FBS. A no-YnPalm control was incubated in DMSO only in F12 media with 3% FBS. The Cells were washed twice with PBS before lysis in 0.1 ml lysis buffer (1% Triton X-100, 0.1% SDS, PBS, and complete EDTA-free protease inhibitor [Roche]). Total protein concentration of lysates was quantitated using the Pierce BCA assay (Cat. no. 23225) and then adjusted to 1 mg/ml with lysis buffer. For each click reaction, 25 $\mu$g total protein was incubated with click-mix comprising (at final concentration) 100 $\mu$M capture reagent AzTB (azido-TAMRA-biotin), 1 mM $CuSO_4$, 1 mM Tris(2-carboxyethyl)phosphine (TCEP), and 100 $\mu$M Tris[(1-benzyl-1$H$-1,2,3-triazol-4-yl)methyl]amine (TBTA) for 60 min at room temperature. Proteins were then precipitated with two volumes methanol, 0.5 volume chloroform, and one volume water. The protein pellet was washed twice with methanol before boiling in 1× non-reducing loading buffer for 5 min and loading onto a 4–15% TGX stain-free gel (4568085; Bio-Rad). After SDS–PAGE, the gels were visualised using a Typhoon FLA 9500 imager (excitation laser: 532 nm, emission filter: LPG [575–700 nm]; GE Healthcare) to detect TAMRA (i.e., proteins labelled by the click reaction) and Cy5 (for MW markers). Proteins were then transferred onto PVDF for Western blotting using anti-HA (ab18181, 1:1,000; Abcam) and anti-HSP90 (sc-69703; Santa Cruz Biotechnology). Densitometric quantification of TAMRA fluorescence (from Typhoon images) and HA (following ECL) was performed on 16-bit images using Image J 1.47v.

### Acyl–biotin exchange and gel shift analysis of S-palmitoylated IFITM3

A549 cells grown in F12 media containing 10% FBS ($7 \times 10^5$ cells/10 cm dish) were washed twice in PBS and then lysed in 0.5 ml lysis buffer (4% [wt/vol] SDS in triethanolamine [TEA] buffer [pH 7.3, 50 mM TEA, and 150 mM NaCl] containing 5 mM EDTA and complete EDTA-free protease inhibitor [Roche]). The total protein concentration of lysates was quantitated using the Pierce BCA assay. 75 $\mu$g of total protein in 100 $\mu$l of lysis buffer was subjected to acyl–biotin exchange using a modified protocol (30) (Fig S3A) in which, after hydroxylamine ($NH_2OH$) cleavage, alkylation was performed with biotin–maleimide (B1267; Sigma-Aldrich) in TEA buffer at 1 mM final concentration for 2 h at room temperature. After methanol–chloroform–$H_2O$ precipitation (2:0.5:1), protein pellets were resuspended in 4× non-reducing SDS–PAGE buffer (8% SDS, 40% glycerol, 0.24M Tris HCl, pH 6.8, and 0.02% bromophenol blue), heated for 5 min at 95°C and allowed to cool for 30 min 6 $\mu$l (=10 $\mu$g) total protein was then reacted with streptavidin (S4762; Sigma-Aldrich) in PBS (total reaction volume 24 $\mu$l) for 5 min at RT. Assuming that IFITM3 comprised all of the total protein, streptavidin was used at a 0.5× molar excess and titrations showed streptavidin was non-limiting at this amount (Fig S3B). The samples were then run on 4–15% TGX SDS–PAGE gels, with cooling packs to ensure heat generation did not disrupt the biotin–streptavidin interaction. Proteins were transferred to PVDF membranes, blocked in 5% BSA, and then blotted using anti-HA (ab18181; Abcam, 1:1,000 in 0.5% BSA). After ECL detection of HA, the membranes were washed in TBS-T, blocked again in 5% BSA overnight, and re-blotted using anti-actin (ab6276; Abcam, 1:1,000 in 0.5% BSA) to provide a loading control. Image J was used to quantitate HA signal density from ECL images using lane intensity plots. The proportion of total IFITM3 that was palmitoylated (and therefore shifted by streptavidin binding to biotin-modified IFITM3-HA) was calculated as (HA signal ≥ 75 kDa)/(total HA signal in lane).

### Western blotting

Unless otherwise stated, the cells were lysed in Triton X-100 lysis buffer (1% [vol/vol] Triton X-100, 25 mM Tris–HCl, pH 7.5, 100 mM NaCl, 1 mM EDTA, and 1 mM EGTA containing complete EDTA-free protease inhibitor). Protein concentrations were determined using the Pierce BCA assay. Proteins were separated by 15% SDS–PAGE and transferred to PVDF membranes. Membranes were blocked in 5% milk in PBST (PBS with 0.1% [vol/vol] Tween 20) and incubated with antibodies diluted in PBST, either for 1 h at RT or overnight at 4°C. The following primary antibodies were used: rat anti-HA (clone 3F10, 1:1,000; Roche), mouse anti-HA (ab18181, 1:1,000; Abcam), rabbit anti-calreticulin (PA3-900, 1:5,000; Thermo Fisher Scientific), and anti-calnexin (CANX) (ab22595, 1:1,000; Abcam). For blots imaged using an Odyssey system (Li-COR), goat antirat IRDye 800 secondary (1:10,000) or goat antirabbit IRDye 680 (1:10,000) was used. For ECL

detection, HRP-conjugated secondary antibodies and Luminata HRP substrate (WBLUR0100; Millipore) were used.

## Supplementary Information

## Acknowledgements

CTO Benfield was supported by a Leverhulme International Academic Fellowship (IAF-2014-021) from The Leverhulme Trust. F MacKenzie, M Mazzon, S Weston, and M Marsh are supported by the UK Medical Research Council (MRC) funding to the MRC-UCL Laboratory for Molecular Cell Biology University Unit (MC_UU00012/1 and MC_U12266B). EC Holmes is supported by an Australian Research Council Australian Laureate Fellowship (FL170100022). This work was supported by Cancer Research UK (Programme Foundation Award C29637/A20183 to EW Tate) and the Seventh Framework Programme of the European Union (PIEF-GA-2013-623648 to M Ritzefeld).

### Author Contributions

CTO Benfield: conceptualization, resources, data curation, formal analysis, funding acquisition, investigation, methodology, project administration, and writing—original draft, review, and editing.
F Mackenzie: conceptualization, resources, data curation, formal analysis, validation, investigation, visualization, methodology, project administration, and writing—original draft, review, and editing.
M Ritzefeld: conceptualization, resources, formal analysis, supervision, funding acquisition, investigation, methodology, project administration, and writing—original draft, review, and editing.
M Mazzon: conceptualization, resources, data curation, formal analysis, supervision, funding acquisition, investigation, methodology, project administration, and writing—review and editing.
S Weston: resources and investigation.
EW Tate: resources, data curation, formal analysis, funding acquisition, validation, investigation, visualization, methodology, project administration, and writing—original draft, review, and editing.
BH Teo: conceptualization, resources, software, formal analysis, supervision, validation, investigation, and writing—review and editing.
SE Smith: resources, supervision, investigation, project administration, and writing—review and editing.
P Kellam: resources, supervision, validation, investigation, project administration, and writing—review and editing.
EC Holmes: conceptualization, resources, software, formal analysis, supervision, funding acquisition, investigation, methodology, and writing—original draft, review, and editing.
M Marsh: conceptualization, resources, formal analysis, supervision, funding acquisition, investigation, methodology, project administration, and writing—review and editing.

### Conflict of Interest Statement

The authors declare that they have no conflict of interest. EW Tate is the director and a shareholder in Myricx Pharma Ltd.

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
