## [Reviewer comments · Life Science Alliance]

Life Science Alliance

Bat IFITM3 restriction depends on S-palmitoylation and a polymorphic site within the CD225 domain

Camilla Benfield, Farrell Mackenzie, Markus Ritzefeld, Michela Mazzon, Stuart Weston, Edward Tate, Boon Han Teo, Sarah Smith, Paul Kellam, Edward Holmes, and Mark Marsh

DOI: <https://doi.org/10.26508/lsa.201900542>

Corresponding author(s): Camilla Benfield, Royal Veterinary College

Review Timeline:

Submission Date:	2019-09-04
Editorial Decision:	2019-09-19
Revision Received:	2019-11-08
Editorial Decision:	2019-11-21
Revision Received:	2019-11-27
Accepted:	2019-11-28

Scientific Editor: Andrea Leibfried

Transaction Report:

September 19, 2019

Re: Life Science Alliance manuscript #LSA-2019-00542-T

Dr. Camilla Tamsin Olivia Benfield
Royal Veterinary College
Hawkshead Lane
Hatfield, Hertfordshire AL9 7TA
United Kingdom

Dear Dr. Benfield,

Thank you for submitting your manuscript entitled "Comparative analysis reveals adaptive evolution of bat IFITMs and a novel antiviral determinant" to Life Science Alliance. The manuscript was assessed by expert reviewers, whose comments are appended to this letter.

As you will see, the reviewers appreciate your data and support publication of a slightly revised version. We would thus like to invite you to submit a revised version of your manuscript to us, addressing the few points raised by the reviewers. Point 2 of reviewer #3 can get addressed by discussion.

Thank you for this interesting contribution to Life Science Alliance. We are looking forward to receiving your revised manuscript.

Sincerely,

Andrea Leibfried, PhD

Executive Editor
Life Science Alliance
Meyerhofstr. 1
69117 Heidelberg, Germany
t +49 6221 8891 502
e a.leibfried@life-science-alliance.org
www.life-science-alliance.org

B. MANUSCRIPT ORGANIZATION AND FORMATTING:

Reviewer #1 (Comments to the Authors (Required)):

The manuscript by Camilla and coworkers describes the phylogenetic and functional analysis IFITM3 in bats. This study is significant since bats are natural hosts to zoonotic and emerging viruses, and interesting setting for virus-host co-evolution. IFITM3 has emerged as key IFN-effector that prevents virus entry in host cells. Interestingly, the authors identified P70 in IFITM3 CD225 domain as a highly variable but functionally important site. They show that IFITM3 P70W mutant

has impaired antiviral activity and altered subcellular localization. Interestingly, the P70W mutant shows significant loss in S-palmitoylation. They further characterized the palmitoylation level, antiviral activity and subcellular localization of single, double and triple IFITM3 Cys mutants and show that S-palmitoylation also regulates the antiviral activity of bat IFITM3. This has been previously shown for mice and human and highlights the conserved regulatory role of IFITM3 S-palmitoylation as well as how neighboring mutations impacts fatty-acylation. This is an important and well executed study. I am pleased to recommend publication with some minor revisions listed below.

- 1) The authors should comment on the differences in wt and P70W protein expression levels, which may also be key determinant of antiviral activity.
- 2) The sub-cellular localization of wt IFITM3 and P70W could be improved by co-staining with plasma membrane and endolysosomal markers.

Reviewer #2 (Comments to the Authors (Required)):

The authors performed a comparative analysis of IFITM genes of mammals including many bat species and they show an adaptive evolution of IFITMs in bats. They study the antiviral activity of these IFITMs against ZIKV, Lyssavirus, SFV and IAV. They describe a novel amino acid variation potentially involved in IFITM function: S-palmitoylation at position 70 is critical for Chiropteran IFITM3 localization and function. The article is well written and there are no major technical issues.

Minor points:

1. Figures 1, 5 and 6 : Why is WT human IFITM expressed at a lower level than all other IFITM mutants ? Is this simply due to transduction variability or are these mutants more stable/expressed?
2. Sup. Figure 1 B : It would be nice to include the mean fluorescence intensity data of the flow cytometry panel in addition to the percentage of positive cells.
3. Is it possible to perform silencing experiments of endogenous IFITM in bat cells? This should be at least discussed

Reviewer #3 (Comments to the Authors (Required)):

Manuscript Nr: LSA-2019-00542-T

Benfield et al., "Comparative analysis reveals adaptive evolution of bat IFITMs and a novel antiviral determinant"

The authors investigated bat IFITM3 variants against infections by Zika (ZIKV), Semliki Forest (SFV)

and influenza A virus (IAV). Within mammalian IFITM proteins amino acid position 70 is occupied with the amino acids P, T, V, A, G or W, suggesting selection pressure on this position. The authors demonstrate that bat IFITM3 loses restriction of the three tested viruses when position 70 is mutated from P to W. This lack of viral restriction was associated with diminished trafficking to the cell membrane for internalization and decreased palmitoylation as posttranslational modification. Therefore, the authors tested if mutants in the palmitoylation sites aa71, 72 and 105 demonstrated similar features. Indeed, especially the C72A mutant of bat IFITM3 lost restriction of the three tested viruses and had the most pronounced perinuclear localization. From these data the authors conclude that the polymorphism on position 70 affects the antiviral function of IFITM proteins by influencing palmitoyl mediated trafficking to the cell membrane for viral restriction after internalization.

These are interesting data on the influence of IFITM polymorphisms on localization of these antiviral proteins and the requirement of palmitoylation on their antiviral function. However, it remains unclear if this behavior would also be conserved in bat cells and what the evolutionary pressure was to generate IFITM proteins with different amino acids in position 70.

Major comments:

1. The authors have tested all of their IFITM3 mutants in human A549 lung epithelial cells. As they point out themselves the polymorphism in position 70 was better tolerated by human than bat IFITM3. This might be different in a bat cell line. Therefore, the authors should provide some information on the trafficking of bat IFITM3 P70W in bat cells.
2. The authors speculate in their discussion that IFITM3 functions during pregnancy might put evolutionary pressure on this protein not to become too efficient for virus restriction. Can they find any cell biological evidence for this? Does overexpression of the most efficient IFITM3 variant against ZIKV, SFV and IAV compromise cellular functions or even decrease cell survival?
3. Alternatively the differences in cellular localization could indicate different functions of IFITM3 at different stages of viral life cycles. While palmitoylation seems required for localization to the cell membrane and restriction of virus entry, perinuclear localization might compromise replication at later timepoints. Does ZIKV, SFV and IAV replication get inhibited at later stages of viral replication by the bat IFITM3 P70W mutant, maybe even better than by cell membrane targeted IFITM3?

Minor comments:

1. The title could include that palmitoylation efficiency determines antiviral function of IFITM3. In its current form it does not indicate the main result of the manuscript.

In summary, the study defines palmitoylation, dependent on a polymorphic sequence in the IFITM genes, as an important characteristic for the anti-viral function of this protein. However, it should be confirmed that this holds true in additional cellular backgrounds and some information on the basis for the observed polymorphism should be provided.

Dear Dr Leibfried

Thank you for your editorial decision letter of 19th September regarding our manuscript entitled "*Comparative analysis reveals adaptive evolution of bat IFITM3 and a novel antiviral determinant*" (#LSA-2019-00542-T).

We appreciate the reviewers' comments and are pleased that they support publication of a revised version to *Life Science Alliance* that addresses the few points of concern. Below we have appended the reviewers' full comments and respond to each of the points, highlighting where we have revised our manuscript accordingly.

Reviewer #1 (Comments to the Authors):

The manuscript by Camilla and coworkers describes the phylogenetic and functional analysis IFITM3 in bats. This study is significant since bats are natural hosts to zoonotic and emerging viruses, and interesting setting for virus-host co-evolution. IFITM3 has emerged as key IFN-effector that prevents virus entry in host cells. Interestingly, the authors identified P70 in IFITM3 CD225 domain as a highly variable but functionally important site. They show that IFITM3 P70W mutant has impaired antiviral activity and altered subcellular localization. Interestingly, the P70W mutant shows significant loss in S-palmitoylation. They further characterized the palmitoylation level, antiviral activity and subcellular localization of single, double and triple IFITM3 Cys mutants and show that S-palmitoylation also regulates the antiviral activity of bat IFITM3. This has been previously shown for mice and human and highlights the conserved regulatory role of IFITM3 S-palmitoylation as well as how neighboring mutations impacts fatty-acylation.

This is an important and well executed study. I am pleased to recommend publication with some minor revisions listed below.

1) The authors should comment on the differences in wt and P70W protein expression levels, which may also be key determinant of antiviral activity.

Response: We agree with this statement and in our original submission we provided western blot panels showing IFITM3 expression levels in the same cells used to analyse virus infectivity (Figs 3A, 3B, 7). Furthermore, we provided quantitative analysis of expression of wt, P70W and all other codon 70 point mutants of IFITM3 in Supplementary Fig S1, using replicate samples and two techniques: western blotting and flow cytometry. We therefore concluded, as stated in our original submission, that '*Expression levels of wt mbIFITM3 and mbIFITM3 P70W were comparable when analyzed by western blotting (Fig. 3A, 3B, S1A) and flow cytometry (Fig. S1B)*' [lines 226-228]. In our revised manuscript we have added the

following sentence: '*Thus, differences in expression do not account for the different antiviral activities of these proteins*' [lines 228-229] to reinforce this important point. In the case of human IFITM3, the P70W substitution did not change protein expression levels or antiviral activity.

2) The sub-cellular localization of wt IFITM3 and P70W could be improved by co-staining with plasma membrane and endolysosomal markers.

Response: In our original submission, Fig 4 included immunofluorescence images of non-permeabilised cells labelled with anti-HA antibody to detect wt and P70W human and microbat IFITM3-HA; these panels were labelled 'intact'. In the absence of detergent, the HA label on these intact cells can only be located at the cell surface, shown both for intact fixed cells (Fig 4A) and intact live cells (Fig 4B). Therefore, we feel the data presented already address the issue of plasma membrane localisation of wt IFITM3 and P70W, as raised by Reviewer 1.

In response to the comment that the sub-cellular localization of wt IFITM3 and P70W could be improved by co-staining with endolysosomal markers, we have performed additional experiments in which we co-stained cells for IFITM3-HA and CD63, a marker for late endosomes and multi-vesicular bodies that co-localises with wt human and microbat IFITM3 (Amini-Bavil-Olyaei *et al.*, 2013; Benfield *et al.*, 2015; Weston *et al.*, 2016; Feeley *et al.*, 2011; Huang *et al.*, 2011; Jia *et al.*, 2012; Lu *et al.*, 2011), and now include these data in panel C of Supplementary FigS2. This figure shows that CD63-positive vesicles are quite widely dispersed within cells, and while some co-localisation of mbIFITM3 P70W with CD63 is evident, this is much less marked than the clear overlap seen between the perinuclear IFITM3-HA staining for P70W and cysteine mbIFITM3 mutants and the Golgi markers Giantin and TGN46 (Fig 4C, 8B, S2A, S2B). We have added the statement 'perinuclear IFITM3 co-localized.... and to a much lesser extent with CD63-positive endolysosomes distributed throughout the cells (Fig. S2C)' [lines 368-369]. We have kept this comment brief so as not to detract from the key message that the S-palmitoylation-deficient mutants of mbIFITM3 co-localise with Golgi markers at perinuclear sites.

Reviewer #2 (Comments to the Authors):

The authors performed a comparative analysis of IFITM genes of mammals including many bat species and they show an adaptive evolution of IFITMs in bats. They study the antiviral activity of these IFITMs against ZIKV, Lyssavirus, SFV and IAV. They describe a novel amino acid variation potentially involved in IFITM function: S-palmitoylation at position 70 is critical for Chiropteran IFITM3 localization and function. The article is well written and there are no major technical issues.

Minor points:

1. Figures 1, 5 and 6 : Why is WT human IFITM3 expressed at a lower level than all other IFITM3 mutants ? Is this simply due to transduction variability or are these mutants more stable/expressed?

Response: We think the reviewer is referring to the western blot panels in Figure 3 (not Figure 1 which shows phylogenetic analysis) and Figures 5 and 6. Quantitation of anti-HA western blots (Fig S1A) showed the wt human IFITM3 is relatively underexpressed compared to the codon 70 mutants of human IFITM3, although three mutants (P70W, P70G and P70T) had <2 fold expression difference relative to wt which was not statistically significant. There is a similar trend in the per cell expression of wt and mutant human IFITM3-HA analysed by flow cytometry (Fig S1B lower panel), which likely explains the western blot data, and which could be due to variation in lentiviral transduction efficiency when creating the different cell lines. Alternatively, differences in the rate of protein synthesis or turnover might underlie the different steady state IFITM3 protein levels. It is possible that ubiquitination, which affects protein turnover, is influenced by the structure of the proteins determined by residue 70.

By anti-HA western blotting the microbat IFITM3 proteins appear better expressed than wt human IFITM3, and we have seen this previously in independent monoclonal cell lines (Benfield *et al.* 2015, Fig S3). However, the fluorescence intensity of cells expressing microbat IFITM3-HA is actually lower than for human IFITM3-HA when assessed by anti-HA staining and flow cytometric analysis (Fig S1B lower panel). We believe this is because the wt and mutant human IFITM3 cell lines were made in parallel using lentivirus that had been concentrated by centrifugation, which likely led to a greater number of genome integration events per cell than for the microbat IFITM3 wt and P70W cell lines which were made in parallel using unconcentrated lentivirus. In view of these flow cytometry data, as well as the strong cellular expression of wt human IFITM3-HA we observe by immunofluorescence, anti-HA western blotting may underestimate the expression of wt human IFITM3. One reason for this could be cleavage of the HA tag when making the cell lysates for western blotting, for which we have some evidence using anti-human IFITM3 antibodies. These antibodies only poorly recognise the microbat IFITM3 and cannot be used to compare the relative expression of human and microbat IFITM3-HA constructs we express in our cell lines.

However, for our manuscript, the important comparison is between wt microbat IFITM3 and microbat IFITM3 P70W, which show equivalent expression by all analysis methods used (see response to point 1 of reviewer 1 above).

2. Sup. Figure 1 B : It would be nice to include the mean fluorescence intensity data of the flow cytometry panel in addition to the percentage of positive cells.

Response: As requested, we have now added a lower panel to Fig S1B showing the mean HA fluorescence intensity of the IFITM₃-HA positive cells for each cell line (discussed above).

3. Is it possible to perform silencing experiments of endogenous IFITM in bat cells? This should be at least discussed

Response: We have previously performed siRNA silencing of endogenous IFITM₃ in primary microbat (*M. myotis*) cells (Benfield *et al.* 2015, PMID: 2561458; Figure 8) and refer to this in the introduction to our current manuscript [lines 145-146] (*'We previously showed that microbat IFITM₃ ... at normal expression levels in primary microbat cells, inhibits infection by pH-dependent enveloped viruses [39]'*). This background information highlights that the microbat IFITM₃ protein studied here is a bona fide restriction factor in cells from its species of origin. However, to address the aim of the current study of analysing the phenotypic effects of particular IFITM₃ amino acid residues, an over-expression system- as we have used- is appropriate.

Reviewer #3 (Comments to the Authors):

The authors investigated bat IFITM₃ variants against infections by Zika (ZIKV), Semliki Forest (SFV) and influenza A virus (IAV). Within mammalian IFITM proteins amino acid position 70 is occupied with the amino acids P, T, V, A, G or W, suggesting selection pressure on this position. The authors demonstrate that bat IFITM₃ loses restriction of the three tested viruses when position 70 is mutated from P to W. This lack of viral restriction was associated with diminished trafficking to the cell membrane for internalization and decreased palmitoylation as posttranslational modification. Therefore, the authors tested if mutants in the palmitoylation sites aa71, 72 and 105 demonstrated similar features. Indeed, especially the C72A mutant of bat IFITM₃ lost restriction of the three tested viruses and had the most pronounced perinuclear localization. From these data the authors conclude that the polymorphism on position 70 affects the antiviral function of IFITM proteins by influencing palmitoyl mediated trafficking to the cell membrane for viral restriction after internalization.

These are interesting data on the influence of IFITM polymorphisms on localization of these antiviral proteins and the requirement of palmitoylation on their antiviral function. However, it remains unclear if this behavior would also be conserved in bat cells and what the evolutionary pressure was to generate IFITM proteins with different amino acids in position 70.

Major comments:

1. The authors have tested all of their IFITM₃ mutants in human A549 lung epithelial cells. As they point out themselves the polymorphism in position 70 was better tolerated by

human than bat IFITM₃. This might be different in a bat cell line. Therefore, the authors should provide some information on the trafficking of bat IFITM₃ P70W in bat cells.

Response: We chose A549 cells because this line has low baseline expression of endogenous human IFITM proteins (Brass *et al.*, Cell (2019)). Several reports show that IFITM_{1/2/3} can form hetero-oligomers and show altered restriction when co-expressed, as we note in our introduction. Therefore, use of A549 cells minimises the issue of interactions between endogenous cellular IFITMs and the exogenously expressed IFITM₃, and for this reason is common within the IFITM literature.

While we agree with the reviewer that assessing the trafficking of mbIFITM₃ P70W in bat cells would be of interest, there are several practical and biological reasons why we feel these experiments are out-with the scope and time-frame of our current study. First, interactions with endogenous bat IFITMs might confound the results, as explained above. We do not have available a cell line from the bat species whose IFITM protein we have studied (*Myotis myotis*). Moreover, the primary cells used to clone this gene senesce rapidly, are challenging to culture and our stocks are almost exhausted, thus they would not be appropriate for transfection or transduction studies. Furthermore, very few studies and reagents exist for cell biological studies of bats. Co-localisation studies (e.g. with the Golgi apparatus with which mbIFITM₃ P70W co-localises in A549 cells) would likely be difficult and time-consuming in bat cells since the antibodies raised against primate or other mammalian epitopes are unlikely to 'see' bat proteins and any that did would require careful optimisation. In addition, if organelles were to have subtly different distributions in these cells, it would be difficult to draw conclusions.

We refer to the need for future studies of IFITM variants in their species of origin (e.g. in line 502), but consider that these fall beyond the scope of this paper, whose conclusions are firmly supported by the large body of data currently presented.

2. The authors speculate in their discussion that IFITM₃ functions during pregnancy might put evolutionary pressure on this protein not to become too efficient for virus restriction. Can they find any cell biological evidence for this? Does overexpression of the most efficient IFITM₃ variant against ZIKV, SFV and IAV compromise cellular functions or even decrease cell survival?

Response: Reproduction-based selection on IFITM₃ was recently discussed by Kellam & Weiss (Science, 2019) who commented '*The degree of IFITM₃ expression thus requires a delicate balancing act because a high level will provide better protection from infection, whereas a low level will help to protect fetal development.*' We reference this publication in our discussion, then further speculate that reproduction-based selection (as well as pathogen-based selection) might vary in strength between different mammal species and thereby also drive variation among mammalian IFITMs. This is relevant to our paper since

we have identified sequence variation among mammalian IFITMs as well as differential effects of P70W in human and microbat IFITM₃.

The reviewer asks if we have any cell biological evidence for strong IFITM₃-mediated restriction compromising cellular functions. In the course of our research we analysed cellular proliferation (via EdU DNA incorporation) in the IFITM₃-expressing A549 cells used throughout this manuscript. These data have been added to this re-submission as supplementary Fig S4 and the conclusion '*Our initial studies show that huIFITM₃ or mbIFITM₃ wt or codon 70 mutants do not affect proliferation of A549 cells (Fig S4)*' has been added to the discussion (lines 504-505). These data include the most efficient IFITM₃ variant against ZIKV, SFV and IAV, namely human IFITM₃ P70A, in line with the reviewer's request. While the editor advised in her decision letter that this point could be addressed by discussion, these additional data strengthen our response by directly addressing the reviewer's comment. The experimental data are consistent with our general observations during cell culture and microscopic examination that expression of either the wt or different mutant IFITM₃ proteins had no noticeable effect on the growth or survival of A549 cells relative to untransduced A549 cells cultured in parallel.

IFITM₃ has cell-type specific and pleiotropic effects, with reports of both pro- and anti-proliferative effects on tumorigenesis (Min *et al.*, FEBS Open Bio. 8, 1299 (2018); Gomez-Herranz *et al.*, Cell Signal, 60, 39 (2019); Alteber *et al.*, Immunol. Cell Biol. 96, 284 (2018)). Similarly, the recently described effect of IFITM₃ to block syncytin-mediated fusion is likely only one aspect of a complex role of IFITM₃ in embryonic development. Were there a selective pressure to reduce IFITM₃ efficiency, by downregulating expression levels or expressing a less active splice variant for example, these effects might occur only locally or temporally in placental tissues, the major site of syncytin expression.

3. Alternatively the differences in cellular localization could indicate different functions of IFITM₃ at different stages of viral life cycles. While palmitoylation seems required for localization to the cell membrane and restriction of virus entry, perinuclear localization might compromise replication at later timepoints. Does ZIKV, SFV and IAV replication get inhibited at later stages of viral replication by the bat IFITM₃ P70W mutant, maybe even better than by cell membrane targeted IFITM₃?

Response: The reviewer raises an interesting point. Indeed, IFITM₃ is known to restrict post entry steps of the HIV-1 life cycle, reducing virion infectivity and inhibiting viral protein synthesis. In this case, a differentially localised non-S-palmitoylated IFITM₃ sub-population might conceivably mediate post-entry restriction, suggested by Lee *et al.* 2018, although this has not yet been addressed. Inhibition of post-entry replication steps of the viruses we have tested has not to our knowledge been described: IFITM₃ inhibits the early stages of the replication of ZIKV (Savidis *et al.*, Cell Rep, 2016), and various reports indicate that SFV and

IAV virions become entrapped and degraded within the endosomal pathways used for cell entry and thus fail to initiate downstream replication steps (as summarised in the introduction to our manuscript). Thus, we do not expect IFITM₃ to inhibit later stages of ZIKV, SFV and IAV replication, but as yet we cannot exclude unreported effects or differences in the mechanism of restriction by human and microbat IFITM₃. In our paper, the ZIKV, SFV and IAV infectivity assays detect inhibition of any step prior to the read-out of viral protein synthesis (Fig 3A-C), while IAV yields, measured as PFU released into the supernatant, captures effects upon the whole replication cycle (Fig 3D). Across all these assays, microbat IFITM₃ P70W was less inhibitory than wt, and in some assays barely provided any restriction compared to untransduced controls (e.g. IAV infectivity, Fig 3C). Our analysis of the Cys mutants indicated that reduced palmitoylation and re-localisation to perinuclear sites underlies this loss of restriction. Therefore, in response to the reviewer's question, we find no evidence to suggest that the microbat IFITM₃ P70W mutant inhibits later stages of viral replication '*maybe even better than by cell membrane targeted IFITM₃*'.

Minor comments:

1. The title could include that palmitoylation efficiency determines antiviral function of IFITM₃. In its current form it does not indicate the main result of the manuscript.

Response: When preparing our initial submission, we cut the reference to palmitoylation from the title due to word limit restrictions. However, we agree that this is an important result and consequently have amended the title to '*Bat IFITM₃ restriction depends on S-palmitoylation and a polymorphic site within the CD225 domain*' (from the previous title '*Comparative analysis reveals adaptive evolution of bat IFITMs and a novel antiviral determinant*').

In summary, the study defines palmitoylation, dependent on a polymorphic sequence in the IFITM genes, as an important characteristic for the anti-viral function of this protein. However, it should be confirmed that this holds true in additional cellular backgrounds and some information on the basis for the observed polymorphism should be provided.

Response: Above under point 1 of reviewer 3, we have justified, with practical and biological reasons, why we do not feel that repeating experiments in a bat cell background should be required for this publication. The task of uncovering the basis for the naturally-occurring polymorphism is considerable, requiring further mechanistic studies as we have highlighted [lines 501-504].

In summary, the revised manuscript contains edits and additions to the text, as highlighted above, as well as new data to address the reviewers' requests (Fig S4, new panel Fig S1B, Fig S2C). In revision we have also made a few minor edits, which were not requested (shown in our 'with tracked changes' document), to improve and update the text. We have also

followed the editorial formatting guidelines for the journal and hope that, in view of our comprehensive response, you are able to accept our revised manuscript (#LSA-2019-00542-TR) for publication in *Life Science Alliance*. Kindly note that the original submission, and not the revised version, was posted previously on biorxiv.

November 21, 2019

RE: Life Science Alliance Manuscript #LSA-2019-00542-TR

Dr. Camilla Tamsin Olivia Benfield
Royal Veterinary College
Hawkshead Lane
Hatfield, Hertfordshire AL9 7TA
United Kingdom

Dear Dr. Benfield,

Thank you for submitting your revised manuscript entitled "Bat IFITM3 restriction depends on S-palmitoylation and a polymorphic site within the CD225 domain". As you will see, reviewer #3 appreciates the introduced changes and we would thus be happy to publish your paper in Life Science Alliance pending final revisions necessary to meet our formatting guidelines:

- Please make sure that the author order in manuscript and submission system match
- Please link your ORCID iD to your profile in our submission system, you should have received an email with instructions on how to do so
- Please add callouts to figure 7A and 7B to the manuscript text
- Please note that figures, including supplementary figures, can only span a single page, please change for Fig 4, 8, and S2
- Please mention the statistical test used next to the p-values mentioned in the figure legends
- Please introduce more panel descriptors for the figures currently having composite panels (Fig 3, 5, 6, 7, S1)
- Please make sure that there are visible spacers for the IF images shown in Fig S2

A. FINAL FILES:

B. MANUSCRIPT ORGANIZATION AND FORMATTING:

Sincerely,

Reviewer #3 (Comments to the Authors (Required)):

Manuscript Nr: LSA-2019-00542-TR

Benfield et al., "Comparative analysis reveals adaptive evolution of bat IFITMs and a novel antiviral determinant"

The authors investigated bat IFITM3 variants against infections by Zika (ZIKV), Semliki Forest (SFV) and influenza A virus (IAV). Within mammalian IFITM proteins amino acid position 70 is occupied with the amino acids P, T, V, A, G or W, suggesting selection pressure on this position. The authors demonstrate that bat IFITM3 loses restriction of the three tested viruses when position 70 is mutated from P to W. This lack of viral restriction was associated with diminished trafficking to the cell membrane for internalization and decreased palmitoylation as posttranslational modification. Therefore, the authors tested if mutants in the palmitoylation sites aa71, 72 and 105 demonstrated similar features. Indeed, especially the C72A mutant of bat IFITM3 lost restriction of the three tested viruses and had the most pronounced perinuclear localization. From these data the authors conclude that the polymorphism on position 70 affects the antiviral function of IFITM proteins by influencing palmitoyl mediated trafficking to the cell membrane for viral restriction after internalization.

In their revised manuscript version, the authors have responded to all of my concerns. They justify why the IFITM3 wt and mutants cannot be readily tested in bat cells with the current tools available, that their IFITM3 variant expression had no influence on cell proliferation and viability, and that they have not observed any effect of the palmitoylation deficient mutants on viral restriction post entry. They have furthermore clarified the title to include the main finding of their study. These changes have in my opinion significantly improved the manuscript.

Responses to final requests from the Editor

- **Please make sure that the author order in manuscript and submission system match**

Changed so they now match

- **Please link your ORCID iD to your profile in our submission system, you should have received an email with instructions on how to do so**

I have responded to the ORCID email to give permission to Crossref

- **Please add callouts to figure 7A and 7B to the manuscript text**

Since the following findings discussed in the results text pertain to all panels of Fig 7, I have added as below:

'All cysteine mutants showed impaired restriction relative to wt mbIFITM3, with a predominant pattern of greatest restriction by C71A > C105A > P70W > C72A (*Fig. 7A-D*). The double (C71A-C72A) and triple (C71A-C72A-C105A) mutants showed no restriction relative to control cells (*Fig. 7A-D*).'

- **Please note that figures, including supplementary figures, can only span a single page, please change for Fig 4, 8, and S2**

All these 3 figs are now single page figures, having decided this was preferable to splitting the figures. The smallest panels are those in 8B, but at this size still shows the finding of HA and Giantin co-localization at perinuclear sites.

- **Please mention the statistical test used next to the p-values mentioned in the figure legends**

This has been done for all p-values given.

- **Please introduce more panel descriptors for the figures currently having composite panels (Fig 3, 5, 6, 7, S1)**

- Figs 3 and 7- additional panel labels were not added for the western images, as we felt that this was not compatible with the figure layout/labelling and would confuse the figures' messages about antiviral restriction. However to improve figure clarity, blot labels on both figs were amended to read '*anti-HA (IFITM3)*' so it is clear this shows IFITM3 expression for the matched infectivity assays (as described in the legend).
- Fig 5- added panels 'A: Quantification' and 'B: Representative blot'. The fig 5 legend has been amended accordingly and call outs in results text now refer to 5A/5B to direct the reader.
- Fig 6- added panels (A-D) reading either 'quantification' or 'representative blot/gel' for both the acyl exchange and metabolic labelling techniques. The fig 7 legend has been amended accordingly and call outs in results text now refer to specific panels to direct the reader.
- Fig S1- additional panel labels added to bar charts and blot panels. The fig S1 legend has been amended accordingly and call outs in results text now refer to specific panels to direct the reader.

- **Please make sure that there are visible spacers for the IF images shown in Fig S2**

Fig S2 has been reformatted to span a single page and the spacers are visible (please note that sometimes spacers may not be visible on the computer display unless you zoom in). S2 now has panels A and B only and the call outs in the text have been amended accordingly.

November 28, 2019

RE: Life Science Alliance Manuscript #LSA-2019-00542-TRR

Dr. Camilla Tamsin Olivia Benfield
Royal Veterinary College
Hawkshead Lane
Hatfield, Hertfordshire AL9 7TA
United Kingdom

Dear Dr. Benfield,

Thank you for submitting your Research Article entitled "Bat IFITM3 restriction depends on S-palmitoylation and a polymorphic site within the CD225 domain". It is a pleasure to let you know that your manuscript is now accepted for publication in Life Science Alliance. Congratulations on this interesting work.

DISTRIBUTION OF MATERIALS:

Again, congratulations on a very nice paper. I hope you found the review process to be constructive and are pleased with how the manuscript was handled editorially. We look forward to future exciting submissions from your lab.

Sincerely,
